# Interaction of coexposure to inorganic arsenic and manganese: Tight junction injury of the blood–brain barrier and the relationship between oxidative stress and inflammatory cytokines in glial cells

Toshiaki Hitomi☯, Hiroko Okuda☯, Ayako Takata, Hiroshi Yamauchi🆔 *

Department of Preventive Medicine, St. Marianna University School of Medicine, Kawasaki, Japan

☯ These authors contributed equally to this work.
* hyama@marianna-u.ac.jp

## Abstract

Coexposure to inorganic arsenic (iAs) and manganese (Mn) may exacerbate cognitive dysfunction caused by iAs alone. In this study, we investigated the cytotoxicity of coexposure to iAs and Mn in glial cells and the expression and correlation between oxidative stress and inflammatory cytokines. Additionally, we assessed tight junction (TJ) injury using a *rat in vitro* blood-brain barrier (BBB) model. In glial cells, coexposure to iAs and Mn increased cytotoxicity compared to single exposure, suggesting a likely additive effect. iAs exposure significantly increased the expression of antioxidant stress markers, including nuclear factor erythroid 2–related factor 2 (Nrf2) and heme oxygenase-1 (HO-1), relative to Mn exposure. Notably, HO-1 expression was further elevated under coexposure conditions, indicating a potential synergistic effect. Regarding inflammatory cytokines, expression of C-C motif chemokine ligand 2 (MCP-1) and interleukin-6 (IL-6) was slightly higher in the iAs exposure compared to Mn exposure. A synergistic effect was observed in the Mn concentration-dependent increase in IL-6 under coexposure. A significant positive correlation was found between Nrf2 or HO-1 and inflammatory cytokines (MCP-1 and IL-6) ($p < 0.001$), suggesting an interaction between oxidative stress and inflammatory cytokines. The BBB TJ injury was evaluated by measuring the transendothelial electrical resistance values and the Claudin-5 and zonula occludens-1. The results showed expression in iAs exposure but not in Mn exposure. Furthermore, Mn did not affect iAs-induced TJ injury. In conclusion, our findings demonstrate that coexposure to iAs and Mn exerts synergistic effects on oxidative stress and inflammatory cytokines in glial cells. These joint effects may increase the risk of neurotoxicity compared to single-iAs or Mn exposure.

**Data availability statement:** All relevant data are within the manuscript and its Supporting Information files.

**Funding:** This study was financially supported by the Japan Society for the Promotion of Science (JSPS) (https://www.jsps.go.jp) in the form of KAKENHI grants received by AT (JP21H03185 and JP24K22214) and TH (JP21K10422). No additional external funding was received for this study.

**Competing interests:** The authors declare no conflicts of interest.

## Introduction

The global incidence of cognitive dysfunction is increasing, affecting both children and adults. Epidemiological studies increasingly highlight neurotoxic substances, such as inorganic arsenic (iAs) and manganese (Mn), as significant contributors. For instance, iAs exposure, primarily through contaminated groundwater, has been linked to cognitive dysfunction in children across Bangladesh [1–4], India [5,6], China [7,8], and Mexico [9,10]. Similarly, Mn exposure from groundwater and air contamination has been associated with cognitive dysfunction in children from diverse regions, including Bangladesh [11–14], China [15,16], Canada [17,18], Mexico [19,20], the United States [21,22], and Brazil [23,24]. Animal studies have demonstrated that exposure to iAs [25,26] or Mn [27–29] alone can induce cognitive dysfunction. Although the precise mechanisms underlying such cognitive dysfunction remain incompletely understood, accumulating evidence suggests that oxidative stress and proinflammatory cytokines may contribute to the common neurotoxic effects of iAs and Mn [25–29]. Notably, while iAs is strongly neurotoxic, Mn is an essential trace element [30] that becomes neurotoxic only at elevated concentrations. Furthermore, emerging evidence suggests that exposure to either iAs [25,31,33] or Mn [31–33] alone may contribute to the onset and progression of Alzheimer's disease (AD). AD is a prominent neurodegenerative disorder primarily characterized by progressive cognitive decline, typically manifesting with aging in adulthood [34,35]. Despite these findings, a critical gap in our understanding persists regarding the long-term consequences of early-life exposure (i.e., fetal, infant, and childhood periods) to iAs and Mn on the subsequent development and progression of AD.

Previous studies [1–29] have primarily focused on cognitive dysfunction caused by individual exposure to neurotoxicants such as iAs and Mn. However, in real-world environments, coexposure to these toxicants is common, and such combined exposure may induce additive or synergistic neurotoxicity that cannot be predicted from single exposure studies alone. In particular, the mechanisms by which coexposure to iAs and Mn leads to cognitive impairment remain poorly understood, and comparative investigations with single-exposures are urgently needed. For example, in arsenic-contaminated regions of Bangladesh, groundwater is often polluted with both iAs [1–4] and Mn [11–14], and studies suggest that combined exposure may cause more severe cognitive deficits in children than single iAs exposures [36–38]. Although this is often viewed as a localized issue, iAs and Mn co-contamination is a global concern. Groundwater used for drinking and agriculture often contains both of these elements, raising concerns about potential health effects on populations numbering in the millions [39]. In the glacial aquifers of northern United States, high concentrations of iAs and Mn have also been detected, raising concerns about health risks through drinking water [40]. Furthermore, the European Food Safety Authority has expressed concern regarding the dietary intake and neurotoxicity of iAs [41] and Mn [42]. Despite the likelihood that exposure through groundwater and food may result from co-contamination, methodologies for assessing the health risks of coexposure remain underdeveloped. Addressing this gap requires innovative research strategies and analytical approaches. Cognitive dysfunction caused by coexposure to

neurotoxicants should be recognized not merely as a regional issue, but as a broader public health risk affecting populations worldwide.

We are interested in the cognitive dysfunction caused by combined exposure to iAs and Mn and are attempting to clarify the mechanism of its occurrence. Our hypothesis for the mechanism of cognitive dysfunction induced by iAs exposure is as follows: first, oxidative stress induced by iAs triggers tight junction (TJ) injury in the blood–brain barrier (BBB), facilitating the entry of iAs into the brain. This, in turn, triggers oxidative stress and inflammatory cytokine responses in glial cells, leading to neuronal cell damage induced by inflammation and the development of various brain dysfunctions. We used a rat *in vitro* BBB model to confirm this hypothesis and analyze the relationship between iAs exposure and TJ injury in the BBB. We thus confirmed that iAs exposure increases TJ injury in a concentration-dependent manner [43].

Epidemiological studies [11–24] and animal [27–29] experiments have demonstrated that Mn is a neurotoxic substance. Neurodegenerative diseases caused by Mn exposure can be understood as a health effects problem on a larger scale than those caused by iAs exposure. Studies on animals [44,45] and rat *in vitro* BBB models [43] have confirmed that iAs enters the brain through the BBB. The routes by which Mn reaches the brain are the BBB [46], the blood–cerebrospinal fluid barrier (BCB) [47], and the olfactory nerve [48]. Although there are no studies directly comparing these three routes, it is believed that Mn is more likely to reach the brain via active transport through BCB than through the BBB. Both iAs and Mn have routes that the BBB can cross, indicating that the BBB is a potential target for these substances. Obtaining information on the joint effects of iAs and Mn on the BBB is important. If additive or synergistic effects occur between iAs and Mn, TJ injury to the BBB may increase, promoting the transfer of iAs and Mn to brain tissue and increasing their accumulation. Against this background, verifying the coexposure of iAs and Mn compared to their single exposure is important. To specifically address this issue, it is crucial to verify the damage to the BBB caused by coexposure to iAs and Mn using a rat *in vitro* BBB model [43]. Furthermore, it is essential to gather information on the joint effects of oxidative stress and inflammatory cytokines in glial cells and to analyze the results thoroughly. In this study, we evaluated the joint effects of iAs and Mn by distinguishing additive effects—where the combined outcome equals the sum of individual effects ($E_{iAs}$ + $E_{Mn} = E_{iAs + Mn}$)—from synergistic effects, where the combined outcome exceeds this sum ($E_{iAs + Mn} > E_{iAs} + E_{Mn}$). Here, $E_{iAs}$ denotes the effect of exposure to iAs, $E_{Mn}$ the effect of exposure to Mn, and $E_{iAs + Mn}$ the effect of coexposure.

Therefore, this study was conducted with the following objectives: 1) Investigate the cytotoxicity in glial cells induced by single or coexposure to iAs and Mn. 2) Evaluate oxidative stress and inflammatory cytokine responses in glial cells following single or coexposure to iAs and Mn. 3) Examine the TJ injury induced by single or coexposure to iAs and Mn using a rat *in vitro* BBB model. By addressing these objectives, we anticipate clarifying the joint effects of iAs and Mn, particularly concerning their impact on BBB integrity and glial cell function, thereby contributing valuable insights into the mechanisms of coexposure-induced neurotoxicity.

## Materials and methods

### Chemicals

iAs (arsenite; arsenic trioxide, 99.995% purity, CAS No. 1327-53-3) and Mn ($MnCl_2 \cdot 4H_2O$, ≥ 99% purity, CAS No. 13446-34-9) were purchased from Sigma-Aldrich (St. Louis, MO, USA). Stock solutions were prepared using cell culture-grade sterile water (Nacalai Tesque, Inc., Kyoto, Japan) and brain capillary endothelial cell culture medium (PharmaCo-Cell Company Ltd, Nagasaki, Japan). Diluted solutions were used for subsequent experiments.

### Rat *in vitro* BBB model and sample collection

A rat *in vitro* BBB model comprising three cell types (RBT-24H) was purchased from PharmaCo-Cell Company Ltd. [49,50]. This model is based on a triple coculture of primary brain capillary endothelial cells, pericytes, and astrocytes isolated from Wistar rats. The rat *in vitro* BBB model structure consists of brain capillary endothelial cells on the top of a transwell filter in the insert portion, pericytes on the bottom, and astrocytes attached to the bottom of a 24-well plate. The

maturation of the rat *in vitro* BBB model was determined by measuring the transendothelial electrical resistance (TEER) using the Millicell Electrical Resistance System (ERS)-2 Voltohmmeter (Merck Millipore, Billerica, MA, USA). Cells were incubated at 37°C according to the manufacturer's protocol until the TEER value reached ≥150 $\Omega \times cm^2$, which indicates that the TJs are functioning properly [49,50].

The exposure concentrations were determined with reference to survey results from Bangladesh. In a survey on cognitive dysfunction in children, the concentrations of arsenic and Mn in groundwater in an area of chronic arsenic poisoning were reported as 117.8 ± 145.2 µg/L and 1,386 ± 927 µg/L, respectively [1]. For reference, 5 µM iAs corresponds to 378 µg/L, and 10 µM Mn corresponds to 549 µg/L. In this experiment, the lowest exposure concentrations (5 µM iAs and 10 µM Mn) were approximately 3-fold higher than the survey value for iAs and approximately 0.4-fold (i.e., 40%) of the survey value for Mn. Therefore, these concentrations were considered to reflect environmentally relevant exposure levels. Furthermore, in the Mn group, concentrations corresponding to 10-fold and 20-fold the measured value of 1,386 µg/L reported in the survey were also applied. As the experimental conditions, iAs alone (10 µM), Mn alone (10, 100, and 200 µM), and coexposure (iAs + Mn: 10 + 10, 10 + 100, and 10 + 200 µM) were applied in the luminal compartment (i.e., the blood side) for 24 h. The detailed procedures for BBB culture, treatment, and cell sample collection (primary brain capillary endothelial cells, pericytes, and astrocytes) were performed exactly as described in our previous study [43]. Next, we obtained the results of nuclear factor erythroid 2-related factor 2 (Nrf2), heme oxygenase-1 (HO-1), claudin-5, and zonula occludens-1 (ZO-1) in the western blot (WB) analysis related to the evaluation of oxidative stress and TJ injury in the BBB. Furthermore, the expression and localization of claudin-5 and ZO-1 were examined by immunocytochemistry to assess morphological TJ injury in the BBB.

## Rat cerebral cortical mixed glial cell culture

Rat cerebral cortical mixed glial cells (i.e., glial cells) were purchased from PharmaCo-Cell Company Ltd. and used in the experiments. These glial cells, consisting of primary microglia and astrocytes derived from Wistar rats, originated from the same source as the rats used to construct the rat *in vitro* BBB model. These cells were cultured in brain capillary endothelial cell culture medium supplemented with 5 ng/mL recombinant rat granulocyte-macrophage colony-stimulating factor (GM-CSF; FUJIFILM Wako, Osaka, Japan) at 37°C in a humidified 5% $CO_2$ atmosphere. The culture medium was changed once every 3–4 days.

## Cytotoxicity assay

The cytotoxicity of glial cells after chemical treatment was validated using the water-soluble tetrazolium salt-8 (WST-8) assay with the Cell Counting Kit-8 (CCK-8; Dojindo, Tokyo, Japan). On the first day, cells were seeded at a density of 20,000 cells/well in 100 µL of brain capillary endothelial cell culture medium in 96-well plates and incubated overnight at 37°C in a humidified 5% $CO_2$ atmosphere. The next day, the cells were treated with iAs alone (1 and 5 µM), Mn alone (10, 100, and 200 µM), or coexposure (iAs + Mn: 5 + 10, 5 + 100, and 5 + 200 µM). After 24 h of treatment, the CCK-8 solution was added to each well and incubated, after which the absorbance was measured at 450 nm using a microplate luminometer ARVOx4 2030 Multilabel Reader (Perkin Elmer, Waltham, MA, USA).

## Real-time quantitative PCR

After the treatment of glial cells with iAs alone (5 µM), Mn alone (10, 100, and 200 µM), or coexposure (iAs + Mn: 5 + 10, 5 + 100, and 5 + 200 µM) for 24 h, total RNA was extracted using the FastGene™ RNA Premium kit (Nippon Genetics, Tokyo, Japan) according to the manufacturer's instructions. Next, cDNA was synthesized using 1 µg of total RNA using the PrimeScript RT Reagent Kit (Takara Bio, Shiga, Japan) according to the manufacturer's protocol. Real-time quantitative PCR was conducted using the StepOnePlus™ Real-Time PCR System (Thermo Fisher Scientific, Waltham, MA,

USA) with TB Green Premix Ex Taq II (Takara Bio). Primer sets from the Perfect Real-Time support system (Takara Bio, https://www.takara-bio.co.jp/research/prt/) were also used. Oxidative stress-related genes were analyzed using the primer sets targeting Nrf2 (*nuclear factor erythroid 2 (NFE2) like basic leucine zipper (bZIP) transcription factor 2; Nfe2l2*, mRNA; Primer Set ID: RA086678) and HO-1 (*heme oxygenase 1*; *Hmox1*, mRNA; RA086407). Inflammatory cytokines were analyzed using the primer sets targeting MCP-1 (*C-C motif chemokine ligand 2; Ccl2,* mRNA; RA011410), IL-1β (*interleukin 1 beta; Il1b,* mRNA; RA075332) and IL-6 (*interleukin 6; Il6,* mRNA; RA079019). The thermal cycling conditions were as follows: initial activation at 95°C for 30 s, followed by 40 cycles of denaturation at 95°C for 5 s and annealing/extension at 60°C for 30 s. The expression levels of antioxidant stress-related and inflammatory cytokine genes were quantified. Gene expression in each sample was normalized to β-actin (*actin, beta*; *Actb*, mRNA; RA015375) and expressed relative to the control group.

## WB analysis

WB analysis was performed as described previously [43], including sample preparation, gel electrophoresis, membrane transfer, and antibody incubation. The following primary antibodies were used: rabbit monoclonal anti-Nrf2 (1:1000, #33649; Cell Signaling Technology, Inc., Danvers, MA, USA), rabbit polyclonal anti-HO-1 (1:1000, 10701–1-AP; Proteintech Group, Inc., Chicago, IL, USA), mouse monoclonal anti-claudin-5 (1:1000, 35–2500; Thermo Fisher Scientific), rabbit polyclonal anti-ZO-1 (1:500, 61–7300; Thermo Fisher Scientific), and mouse monoclonal anti-β-actin (1:2500, A5316; Sigma-Aldrich). Detection and quantification were performed as described previously [43].

## Immunocytochemistry

The expression of claudin-5 and ZO-1 in vascular endothelial cells of the BBB was evaluated using immunocytochemistry as described previously [43]. Briefly, primary brain capillary endothelial cells on transwell inserts were fixed, permeabilized, and stained with specific primary and secondary antibodies. The following primary antibodies were used: mouse monoclonal anti-claudin-5 (1:100, 35–2500; Thermo Fisher Scientific) and mouse monoclonal anti-ZO-1 (1:100, 33–9100; Thermo Fisher Scientific). DAPI was used for nuclear staining, and fluorescence images were obtained using a BZ-X710 fluorescence microscope (Keyence, Osaka, Japan). The exposure time was standardized to 1/30 s as the imaging condition. After setting a threshold by eye, a line (70 µm) was arbitrarily set such that there were two or more adjacent cells as the region of interest in those images, and then the immunoreactivity of the crossed primary brain capillary endothelial cell membranes was quantitatively measured as a histogram using ImageJ (National Institutes of Health, MD, USA). Stained regions obtained as histogram peaks were averaged in each sample (n = 6, average 40–50 locations) and quantified. The counting method used in this analysis was a modification of the method described previously [51].

## Statistical analyses

Statistical analyses were conducted using IBM SPSS Version 28.0 (IBM Corp, Armonk, NY, USA). All results are summarized as mean ± standard deviation (SD). One-way ANOVAs with Tukey's post hoc tests were used for comparing two or more groups. Spearman's rank correlation coefficient was used to calculate the association between pairs of variables. $p < 0.05$ was considered statistically significant for the two-tailed test.

## Results

### Cytotoxicity, oxidative stress, and inflammatory cytokines in glial cells

**Cytotoxicity of iAs and Mn.** Cytotoxicity in glial cells was evaluated using the CCK-8 method at 24 h after exposure to iAs alone (1 and 5 µM) or Mn alone (10, 100, and 200 µM) or coexposure (iAs + Mn: 5 + 10, 100, and 200 µM) (Fig 1). Although no cytotoxicity was detected in the 1-µM iAs group, the 5-µM iAs group demonstrated significant cytotoxicity,

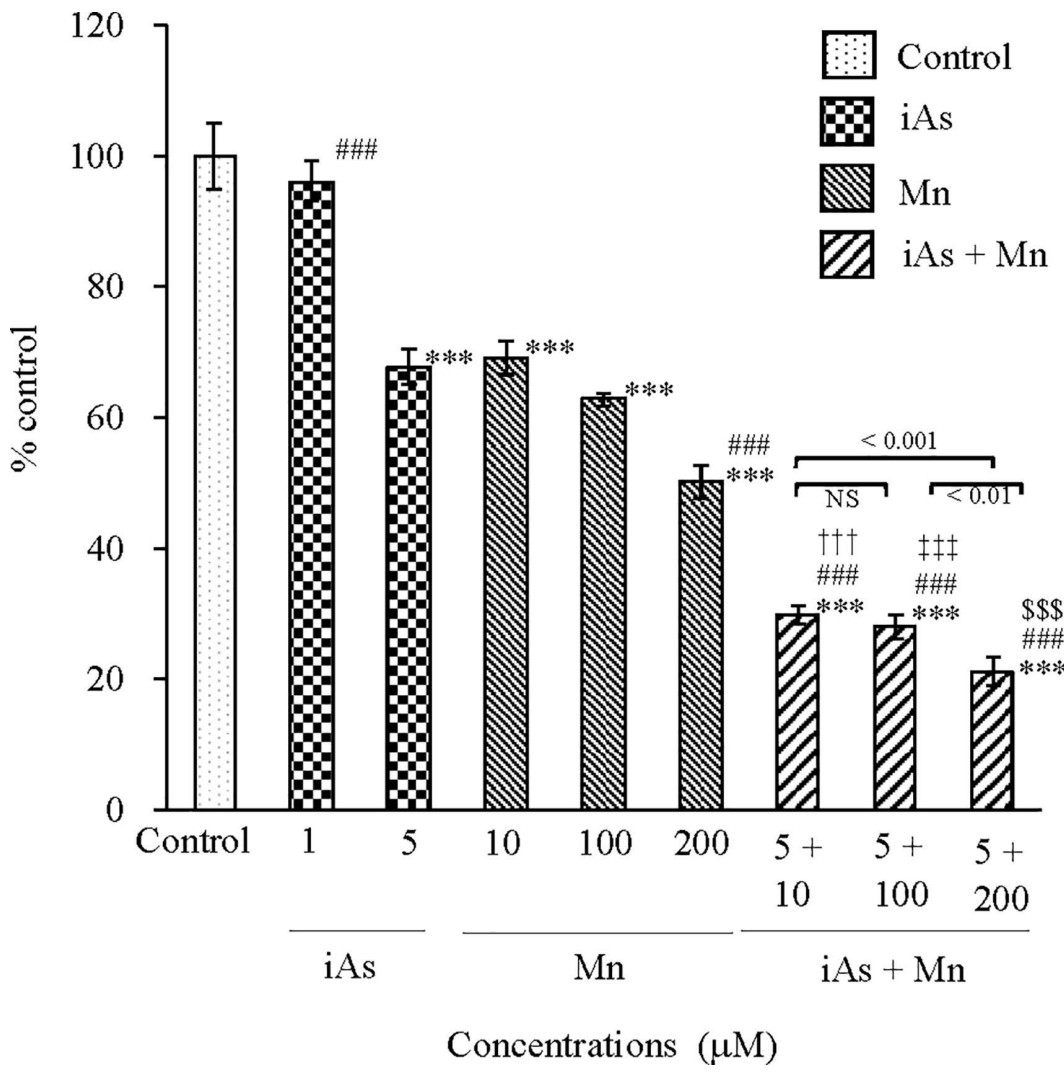

**Fig 1. Cytotoxicity of iAs or Mn alone and coexposure to iAs and Mn in glial cells.** The cytotoxicity of iAs (1 and 5 µM) or Mn (10, 100, and 200 µM) alone and coexposure to (iAs + Mn: 5 + 10, 100, and 200 µM) was evaluated in glial cells using the CCK-8 method 24 h after exposure. Results are expressed as mean ± SD (n = 5). Comparisons of the control group, iAs and Mn alone groups, and iAs and Mn coexposure groups were performed using one-way ANOVA with Tukey's post hoc tests. The levels of statistical significance were as follows: for control, ***$p < 0.001$; for iAs, ###$p < 0.001$. Moreover, the significance levels for each concentration of Mn and the corresponding coexposure were as follows: 10 µM, †††$p < 0.001$; 100 µM, ‡‡‡$p < 0.001$; 200 µM, $$$$p < 0.001$. NS indicates no significant differences among the coexposure groups. Detailed statistical values are provided in S1 Table.

resulting in an approximately 30% reduction in viability compared with that in the control group ($p < 0.001$; see S1 Table). Mn also exerted toxic effects ($p < 0.001$; see S1 Table) on the control group. Furthermore, there were no statistically significant differences in cytotoxicity between the 5-µM iAs and Mn groups. Coexposure to iAs and Mn significantly increased the cytotoxicity compared with that in the control group ($p < 0.001$; see S1 Table). There was an approximately 2.5-fold increase in toxicity compared with exposure to 5 µM iAs ($p < 0.001$; see S1 Table) or Mn (10, 100, and 200 µM) ($p < 0.001$; see S1 Table). This increase in toxicity due to coexposure to iAs and Mn suggested a likely additive effect.

**Oxidative stress induced by iAs and Mn.** In glial cells, real-time PCR analysis was performed to evaluate the mRNA expression levels of the antioxidant stress-related genes *Nrf2* and *HO-1* after exposure to iAs alone (5 µM) or Mn alone

(10, 100, and 200 μM) or coexposure (iAs + Mn: 5 + 10, 100, and 200 μM). The Nrf2 mRNA expression was found to be approximately 1.8-fold higher in the iAs group than in the control group (p < 0.001; see S2 Table, panel A and Fig 2A). In contrast, there was no significant increase in the Mn group (Fig 2A). Coexposure to iAs and Mn resulted in a statistically significant increase in Nrf2 mRNA expression compared with that in the control group (p < 0.001; see S2 Table, panel A), and this pattern of Nrf2 mRNA expression was dependent on Mn concentration (Fig 2A). Similarly, HO-1 mRNA

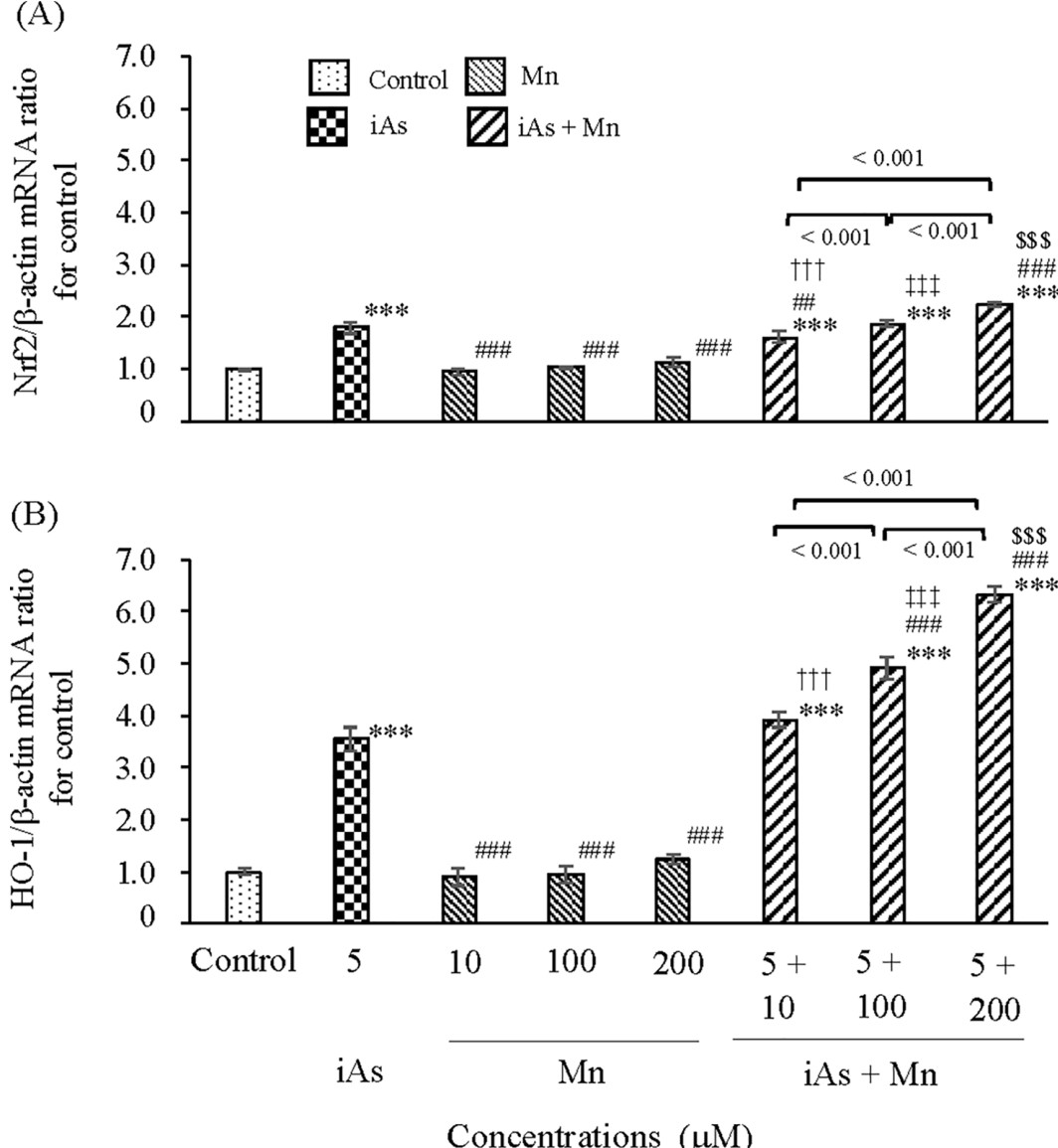

**Fig 2. Oxidative stress of iAs or Mn alone and coexposure to iAs and Mn in glial cells.** iAs alone (5 μM) or Mn alone (10, 100, and 200 μM) or coexposure (iAs + Mn: 5 + 10, 100, and 200 μM) treatment of glial cells for 24 h. After exposure, the expression of antioxidant stress-related genes *Nrf2* (A) and *HO-1* (B) was evaluated by real-time PCR and normalized to β-actin. Data are expressed as Nrf2/β-actin and HO-1/β-actin mRNA ratios. Results are expressed as mean ± SD (n = 4). Comparisons of the control group, iAs and Mn groups, and coexposure to iAs and Mn groups were performed using one-way ANOVA with Tukey's post hoc tests. The levels of statistical significance were as follows: for control, ***p < 0.001; for iAs, ##p < 0.01, ###p < 0.001. Furthermore, the significance levels for each concentration of Mn and the corresponding coexposures were as follows: 10 μM, †††p < 0.001; 100 μM, ‡‡‡p < 0.001; 200 μM, $$$p < 0.001. Detailed statistical values are provided in S2 Table, corresponding to Figs 2A and 2B.

expression was approximately 3.5-fold higher in the iAs group than in the control group (p < 0.001; see S2 Table, panel B), whereas there was no increase in the Mn group (Fig 2B). Coexposure to iAs and Mn significantly increased HO-1 mRNA expression compared to the control group (p < 0.001; see S2 Table, panel B). Furthermore, a synergistic effect of 200 μM-Mn were suggested for the increasing trend of HO-1 mRNA expression (Fig 2B; see S2 Table, panel B). The correlation between Nrf2 and HO-1 mRNA expression among the groups is shown in Table 1. There was a significant correlation in the iAs group (ρ = 0.929, p < 0.001) and the iAs and Mn coexposure group (ρ = 0.809, p < 0.001). There was also a significant correlation in the Mn group (ρ = 0.635, p < 0.01); however, the correlation coefficient was lower than that in the iAs group, suggesting that the effect of Mn on oxidative stress is weaker than that of iAs.

**Inflammatory cytokines induced by iAs and Mn.** The levels of inflammatory cytokines (MCP-1, IL-1β, and IL-6 mRNA) were evaluated by real-time PCR analysis after 24 h (Fig 3). MCP-1 mRNA expression was approximately two-fold higher in the iAs group than in the control group (p < 0.001; see S3 Table, panel A and Fig 3A). However, there was no increase in the Mn group (Fig 3A). Coexposure to iAs and Mn resulted in a significant increase in MCP-1 mRNA expression compared with that in the control group and iAs or Mn alone group (p < 0.001; see S3 Table, panel A and Fig 3A). The IL-1β mRNA expression remained unchanged in the iAs group compared with that in the control group (Fig 3B). The Mn group exhibited a significant difference only at high concentrations (200 μM, p < 0.001; see S3 Table, panel B and Fig 3B). The iAs and Mn coexposure group exhibited a clear trend of elevated IL-1β mRNA expression compared with the control group (p < 0.01, 0.001; see S3 Table, panel B and Fig 3B). The IL-6 mRNA expression was approximately two-fold higher in the iAs group than in the control group (p < 0.001; see S3 Table, panel C and Fig 3C). A slight increase in IL-6 mRNA expression was detected in the Mn group compared with that in the control group; however, a significant difference was observed in the high-concentration groups (100 μM, p < 0.01; 200 μM, p < 0.05; see S3 Table, panel C and Fig 3C). In contrast, the iAs and Mn coexposure group exhibited a significant increase in IL-6 mRNA expression compared with the control group and iAs or Mn alone group (p < 0.001; see S3 Table, panel C and Fig 3C), and this increase was dependent on Mn concentration. A trend toward increased IL-6 mRNA expression suggested an additive effect at 10 μM-Mn, while clear evidence of a synergistic effect was observed at 100 and 200 μM-Mn (Fig 3C; see S3 Table, panel C). Remarkably, regarding the expression of inflammatory cytokines in the iAs and Mn coexposure group, a significant correlation was detected between MCP-1 and IL-6 mRNA expression (ρ = 0.917, p < 0.001) and IL-1β and IL-6 mRNA expression

**Table 1. Spearman correlation coefficients (ρ) among Nrf2 mRNA, HO-1 mRNA, MCP-1 mRNA, IL-1β mRNA, and IL-6 mRNA expression levels.**

|  | Group | Nrf2 | HO-1 | MCP-1 | IL-1β | IL-6 |
|---|---|---|---|---|---|---|
| Nrf2 | iAs | | 0.929*** | 0.714* | 0.905** | 0.929*** |
| | Mn | | 0.635** | 0.479 | 0.662** | 0.753*** |
| | iAs + Mn | | 0.809*** | 0.717*** | 0.737*** | 0.809*** |
| HO-1 | iAs | | | 0.690 | 0.833* | 0.810* |
| | Mn | | | 0.603* | 0.497 | 0.459 |
| | iAs + Mn | | | 0.869*** | 0.871*** | 0.964*** |
| MCP-1 | iAs | | | | 0.762* | 0.690 |
| | Mn | | | | 0.309 | 0.747*** |
| | iAs + Mn | | | | 0.877*** | 0.917*** |
| IL-1β | iAs | | | | | 0.786* |
| | Mn | | | | | 0.747*** |
| | iAs + Mn | | | | | 0.902*** |

iAs group, n = 8; Mn group, n = 16; iAs + Mn group, n = 20.

*p < 0.05; **p < 0.01; ***p < 0.001.

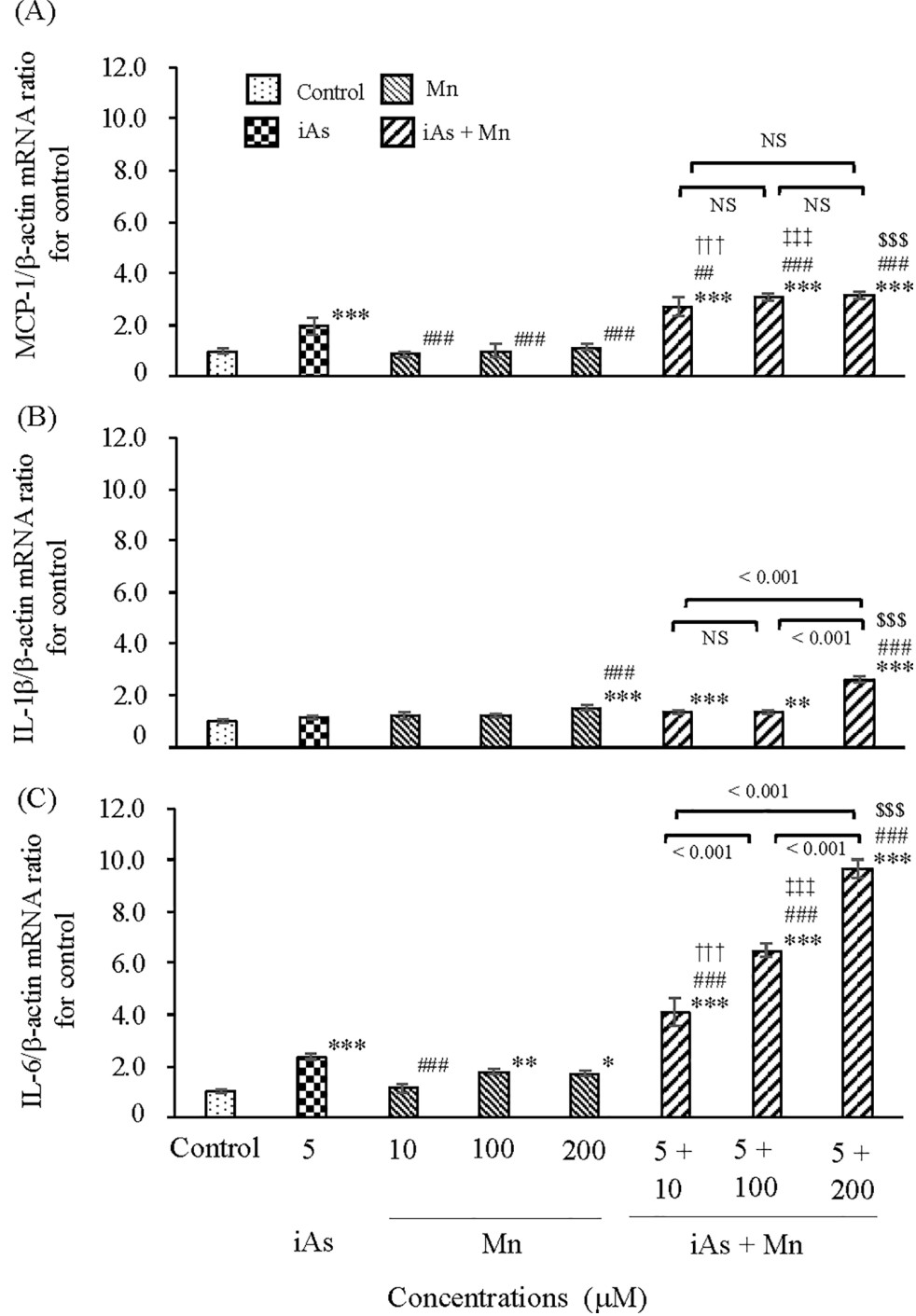

**Fig 3. Inflammatory cytokines in iAs and Mn alone groups and iAs and Mn coexposure groups in glial cells.** iAs alone (5 μM) or Mn alone (10, 100, and 200 μM) or coexposure (iAs + Mn: 5 + 10, 100, and 200 μM) was applied to glial cells for 24 h. mRNA expression levels of the inflammatory cytokines MCP-1 (A), IL-1β(B), and IL-6 (C) were measured by real-time PCR analysis and normalized to β-actin. Data are expressed as MCP-1/β-actin mRNA, IL-1β/β-actin, and IL-6/β-actin ratios. Results are expressed as mean ± SD (n = 4). Comparisons of the control group, iAs and Mn alone groups, and iAs and Mn exposure groups were performed using one-way ANOVA with Tukey's post hoc tests. The levels of statistical significance were as follows: for control, $^{*}p < 0.05$, $^{**}p < 0.01$, $^{***}p < 0.001$; for iAs, $^{\#\#}p < 0.01$, $^{\#\#\#}p < 0.001$. Moreover, the significance levels for each concentration of Mn and the corresponding coexposures were as follows: 10 μM, $^{\dagger\dagger\dagger}p < 0.001$; 100 μM, $^{\ddagger\ddagger\ddagger}p < 0.001$; 200 μM, $^{\$\$\$}p < 0.001$. NS indicates no significant differences among the coexposure groups. Detailed statistical values are provided in S3 Table, corresponding to Figs 3A–3C.

(ρ = 0.902, p < 0.001) (Table 1). This suggests the importance of MCP-1 mRNA or IL-6 mRNA in investigating the levels of inflammatory cytokines.

**Relationship between oxidative stress and inflammatory cytokines.**  We analyzed the relationship between oxidative stress and inflammatory cytokines in the iAs or Mn alone groups and the iAs and Mn coexposure group. The Nrf2 mRNA and HO-1 mRNA expression levels were used as indicators of oxidative stress. As depicted in Fig 2 and S2 Table, iAs exposure induced oxidative stress, whereas Mn exposure did not. The expression of inflammatory cytokines was confirmed in the iAs alone group based on MCP-1 and IL-6 mRNA expression and in the Mn alone group based on IL-1β and IL-6 mRNA expression (Fig 3; see S3 Table). The relationship between oxidative stress and inflammatory cytokines was evaluated using the correlation presented in Table 1. A negative correlation between two indices suggests an inhibitory or antagonistic effect, whereas a positive correlation suggests a synergistic effect. We detected a positive correlation in all the experimental groups. The iAs and Mn coexposure group tended to exhibit a significant correlation compared with the iAs or Mn alone group. In particular, in the iAs and Mn coexposure group, there was a significant positive correlation between HO-1 and MCP-1 mRNA (ρ = 0.869, p < 0.001), HO-1 and IL-1β mRNA (ρ = 0.871, p < 0.001), and HO-1 and IL-6 mRNA (ρ = 0.964, p < 0.001) expression (Table 1).

### Effect of single or combined exposure to iAs and Mn on BBB TJ injury

**Expression of Nrf2 and HO-1 in BBB.**  The rat *in vitro* BBB model was exposed to iAs (10 μM) or Mn (10, 100, and 200 μM) alone or iAs and Mn combination (iAs + Mn: 10 + 10, 100, and 200 μM), after which the protein expression levels of Nrf2 and HO-1 in the vascular endothelial cell and pericyte group at 24 h were evaluated by WB analysis (Fig 4A). Nrf2 protein expression was approximately two-fold higher in the iAs group than in the control group (p < 0.001; see S4 Table, panel A and Fig 4B). However, there was no increase in the Mn group (Fig 4B). The expression of Nrf2 protein in the iAs and Mn coexposure group showed a significant increased trend compared to the control group (p < 0.01, 0.001; see S4 Table, panel A and Fig 4B). However, the values in the coexposure group were at the same level as those in the iAs group, and no additive or synergistic effects were observed (Fig 4B). Furthermore, the expression of HO-1 protein in the iAs, Mn, and iAs+Mn groups showed a trend similar to that of Nrf2 protein expression (p < 0.05, 0.01, 0.001; see S4 Table, panel B and Fig 4C).

Next, we investigated the protein expression of Nrf2 and HO-1 in astrocytes under the same experimental conditions (Fig 5A). Following iAs exposure, Nrf2 protein expression increased to approximately 3-fold that of the control group (p < 0.05; see S5 Table, panel A and Fig 5B). In contrast, no increase in Nrf2 protein expression was observed in the Mn group compared to the control group (Fig 5B; see S5 Table, panel A). Coexposure to iAs and Mn resulted in Nrf2 protein expression compared to the control group (p < 0.001; see S5 Table, panel A) and the iAs group (p < 0.01, 0.001; see S5 Table, panel A). These findings suggest that there may be an additive effect of 10 and 100 μM-Mn on the increase in Nrf2 protein expression in the coexposure of iAs and Mn. Furthermore, a synergistic effect was suggested at 200 μM-Mn. On the other hand, following iAs exposure, the expression of HO-1 protein showed only a slight increase compared to the control group (p < 0.05; see S5 Table, panel B, and Fig 5C). In the Mn group, no significant increase in HO-1 protein expression was observed compared to the control group (see Fig 5C and S5 Table, panel B). Coexposure to iAs and Mn (10, 100, 200 μM-Mn) significantly increased HO-1 protein expression compared with the control and iAs groups (p < 0.001; see S5 Table, panel B). Furthermore, the increased expression of HO-1 protein with coexposure to iAs and Mn suggested a synergistic effect.

The expression of Nrf2 and HO-1 proteins in the BBB was more significant in astrocytes than in vascular endothelial cells and pericytes. In astrocytes exposed to a coexposure of iAs and Mn, the expression of Nrf2 and HO-1 increased, and it was clarified that iAs caused this effect. Additionally, the presence of synergistic effects was suggested (Figs 5B and 5C).

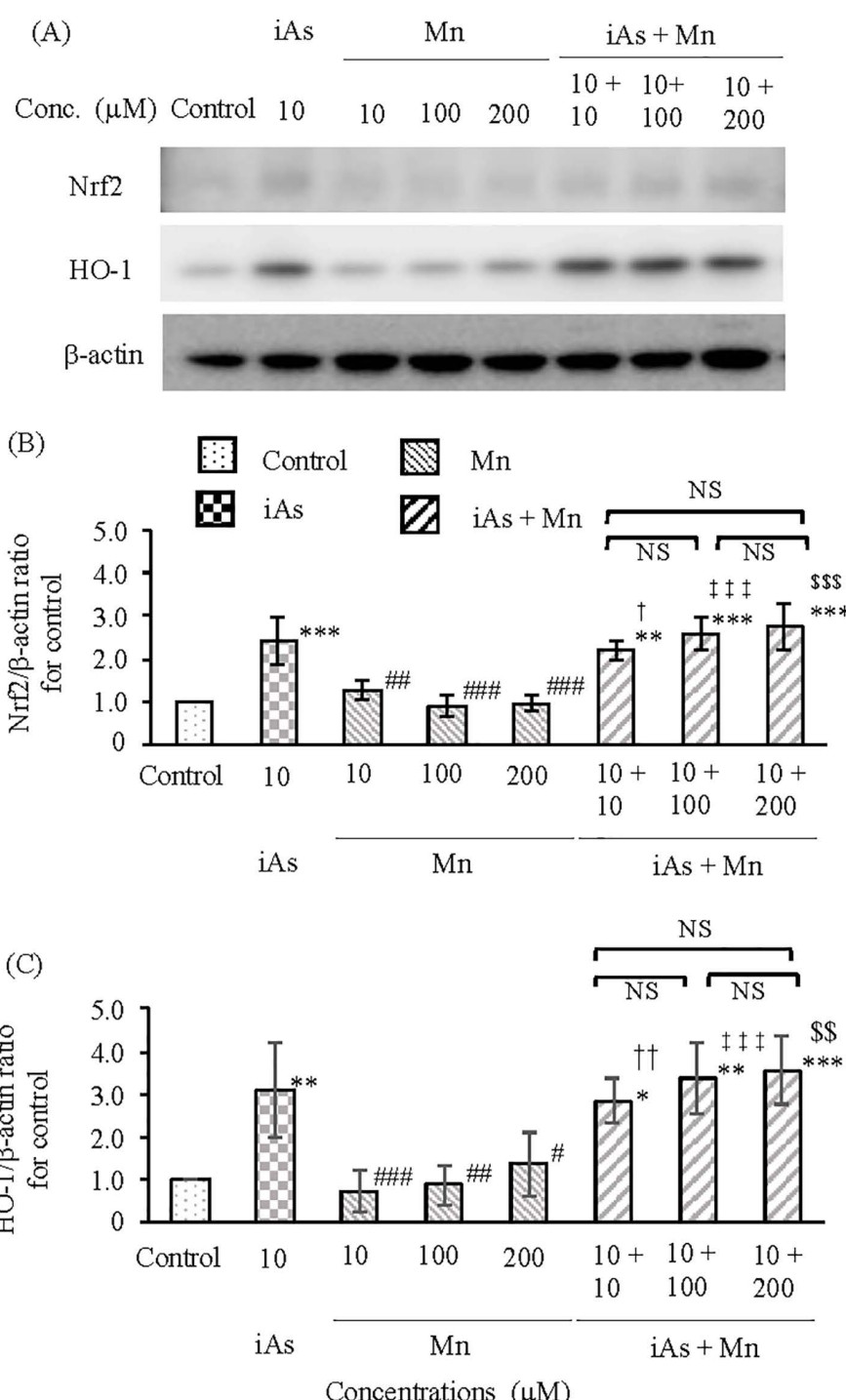

**Fig 4. Expression of Nrf2 and HO-1 proteins in the vascular endothelial cell and pericyte groups after exposure to iAs or Mn alone and coexposure to iAs and Mn. (A)**: Representative western blot images depicting Nrf2 and HO-1 protein expression after exposure to iAs alone (10 μM), Mn alone (10, 100, and 200 μM), and coexposure (iAs + Mn: 10+10, 100, and 200 μM) for 24 **h. (B and C)**: The expression levels of Nrf2 (B) and HO-1 (C) proteins were quantified by densitometry, standardized to β-actin, and compared with the control value of 1. Results are expressed as mean ± SD (n = 4). Comparison of the control group, iAs and Mn alone groups, and iAs and Mn coexposure group was performed using one-way ANOVA with Tukey's post hoc tests. The levels of statistical significance were as follows: for control, $^*p < 0.05$, $^{**}p < 0.01$, $^{***}p < 0.001$; for iAs, $^\#p < 0.05$, $^{\#\#}p < 0.01$, $^{\#\#\#}p < 0.001$. The significance levels for each

concentration of Mn and the corresponding coexposures were as follows: 10 μM, †p < 0.05, ††p < 0.01; 100 μM, ‡‡‡p < 0.001; 200 μM, $$p < 0.01, $$$p < 0.001. NS indicates no significant differences among the coexposure groups. Detailed statistical values are provided in S4 Table, corresponding to Figs 4B and 4C.

**Evaluation of BBB TJ injury by TEER.** TJ injury in BBB can be evaluated by analyzing the changes in TEER values. We evaluated the TJ injury in BBB after exposure to iAs alone (10 μM), Mn alone (10, 100, and 200 μM), and coexposure (iAs + Mn: 10 + 10, 100, and 200 μM) for 24 h using %TEER values (Fig 6). The %TEER was expressed as 100% of the mean control TEER value. The %TEER values in the iAs group decreased by approximately 75% compared with that in the control group (p < 0.001; see S6 Table). In contrast, the Mn group showed no statistically significant decrease in %TEER values. Coexposure to iAs and Mn resulted in a reduction similar in %TEER to that observed after exposure to iAs alone (p < 0.001; see S6 Table). The %TEER values indicate that TJ injury can be attributed to iAs exposure, whereas Mn exposure does not appear to cause such injury. Therefore, it can be concluded that the TJ injury observed in the coexposure group is primarily due to the effect of iAs.

**Evaluation of the TJ proteins claudin-5 and ZO-1.** TJ injury in BBB in the vascular endothelial cell and pericyte groups after exposure to iAs alone (10 μM), Mn alone (10, 100, and 200 μM), and coexposure (iAs + Mn: 10 + 10, 100, and 200 μM) was evaluated by measuring the protein expression patterns of claudin-5 by WB analysis (Fig 7A). The iAs group showed a statistically significant decrease of approximately 60% in claudin-5 expression compared with that in the control group (p < 0.01; see S7 Table, panel A and Fig 7B). In contrast, the Mn groups showed no reduction in claudin-5 expression (Fig 7B; see S7 Table, panel A). The iAs and Mn coexposure group showed a statistically significant decrease in claudin-5 expression compared with that in the control group (p < 0.01; see S7 Table, panel A), a trend that was similar to that in the iAs group (Fig 7B; see S7 Table, panel A). TJ injury in BBB was evaluated by immunofluorescence staining for the expression of claudin-5 proteins (Fig 8). Claudin-5 expression in the control group (Fig 8Aa) was observed as a zonal pattern around the plasma membrane (yellow arrows). In the iAs group, a suppressed (i.e., sparse and weak) expression of claudin-5 was detected in the regions indicated by the red arrowhead (Fig 8Ab), which significantly reduced compared with that in the control group (p < 0.001; see S8 Table and Fig 8B). In contrast, in the Mn groups (Fig 8Ac-e), claudin-5 expression was observed as a banded pattern around the cell membrane, indicated by white arrows, which was not statistically significantly different compared with that in the control group (Fig 8B). In the iAs and Mn coexposure group (Fig 8Af-h), the expression of claudin-5 was observed in the regions indicated by the red arrowheads, which was suppressed compared with that in the control and Mn groups, similar to that in the iAs group (p < 0.001; see S8 Table and Fig 8B).

We also investigated the expression of ZO-1 under the same experimental conditions (Fig 7A) and found a statistically significant decrease of approximately 60% in the iAs group compared with that in the control group (p < 0.001; see S7 Table, panel B and Fig 7C). However, the Mn groups (10 and 100 μM) showed no decrease in ZO-1 expression, but 200 μM-Mn was slightly decreased in the control group (p < 0.05; see S7 Table, panel B and Fig 7C). The iAs and Mn coexposure group showed a statistically significant decrease in ZO-1 expression compared with that in the control group (p < 0.001; see S7 Table, panel B), a trend that was similar to that in the iAs group (Fig 7C; see S7 Table, panel B). The results of fluorescence immunostaining for the expression of ZO-1 proteins are shown in Fig 9. The ZO-1 expression in the control group (Fig 9Aa) was observed as a zonal pattern around the plasma membrane (yellow arrows). In the iAs group, a suppressed (i.e., sparse and weak) expression of ZO-1 was observed in the regions indicated by the red arrowhead (Fig 9Ab), with no statistically significant difference in expression compared with that in the control group (Fig 9B; see S9 Table). In contrast, in the Mn groups (Fig 9Ac-e), the ZO-1 expression was observed as a banded pattern around the cell membrane, indicated by white arrows, and the expression was not statistically significantly different from that in the control group (Fig 9B; see S9 Table). In the iAs and Mn coexposure group (Fig 9Af-h), ZO-1 expression was observed in the regions indicated by red arrowheads and was suppressed compared with that in the control and Mn groups, similar to that in the iAs group (p < 0.001; see S9 Table, Fig 9B).

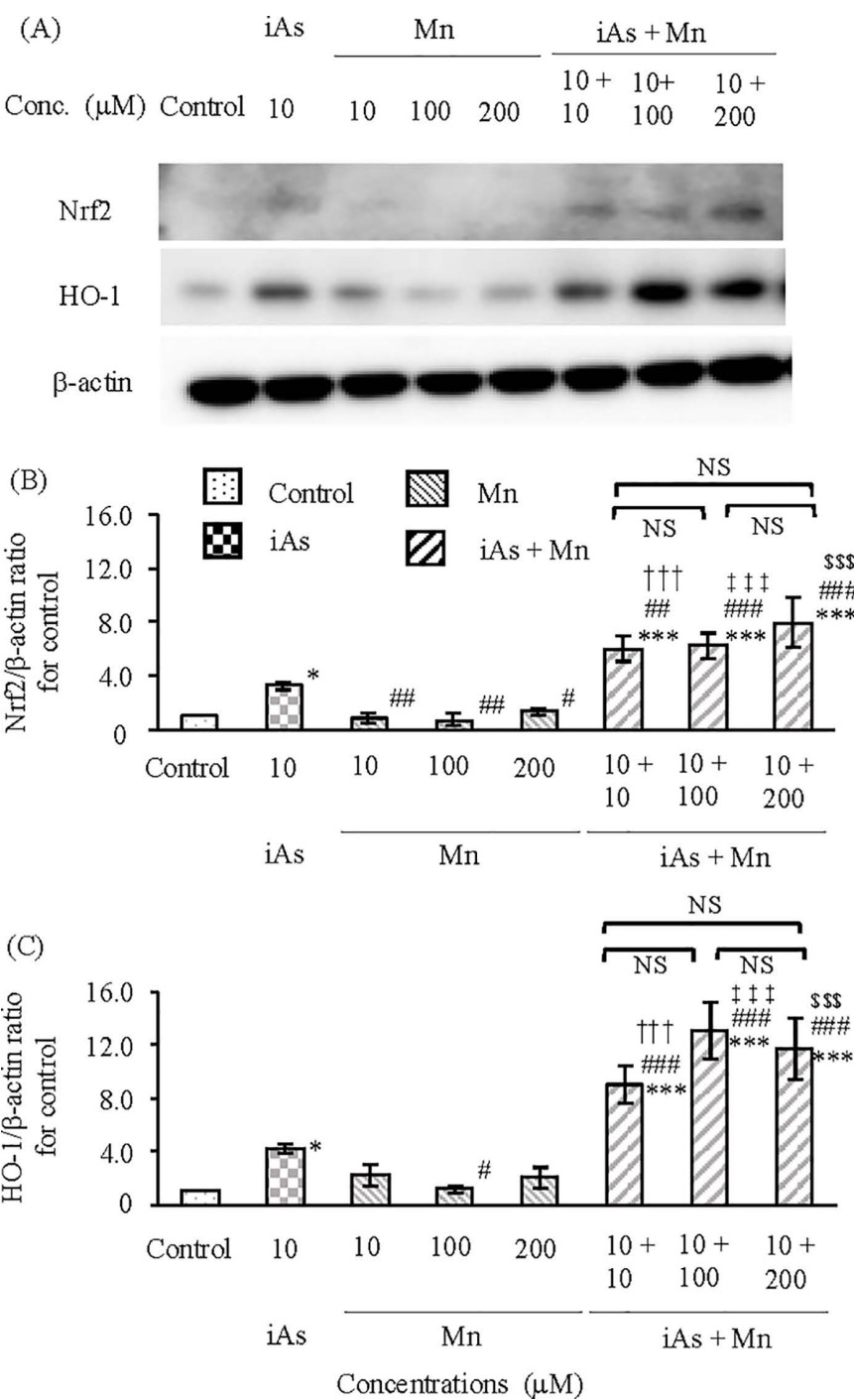

**Fig 5. Expression of Nrf2 and HO-1 proteins in astrocytes after exposure to iAs or Mn alone and coexposure to iAs and Mn. (A)**: Representative western blot images depicting Nrf2 and HO-1 expression after exposure to iAs alone (10 μM), Mn alone (10, 100, and 200 μM), and coexposure (iAs + Mn: 10 + 10, 100, and 200 μM) for 24 **h. (B and C)**: The expression levels of Nrf2 (B) and HO-1 (C) proteins were quantified by densitometry, standardized to β-actin, and compared with the control value of 1. Results are expressed as mean ± SD (n = 4). Comparison of the control group, iAs and Mn alone groups, and iAs and Mn coexposure group was performed using one-way ANOVA with Tukey's post hoc tests. The levels of statistical significance were as follows: for control, $^*p < 0.05$, $^{***}p < 0.001$; for iAs, $^\#p < 0.05$, $^{\#\#}p < 0.01$, $^{\#\#\#}p < 0.001$. The significance levels for each concentration of Mn and the corresponding coexposures were as follows: 10 μM, $^{\dagger\dagger\dagger}p < 0.001$; 100 μM, $^{\ddagger\ddagger\ddagger}p < 0.001$; 200 μM, $^{\$\$\$}p < 0.001$. NS indicates no significant differences among the coexposure groups. Detailed statistical values are provided in S5 Table, corresponding to Figs 5B and 5C.

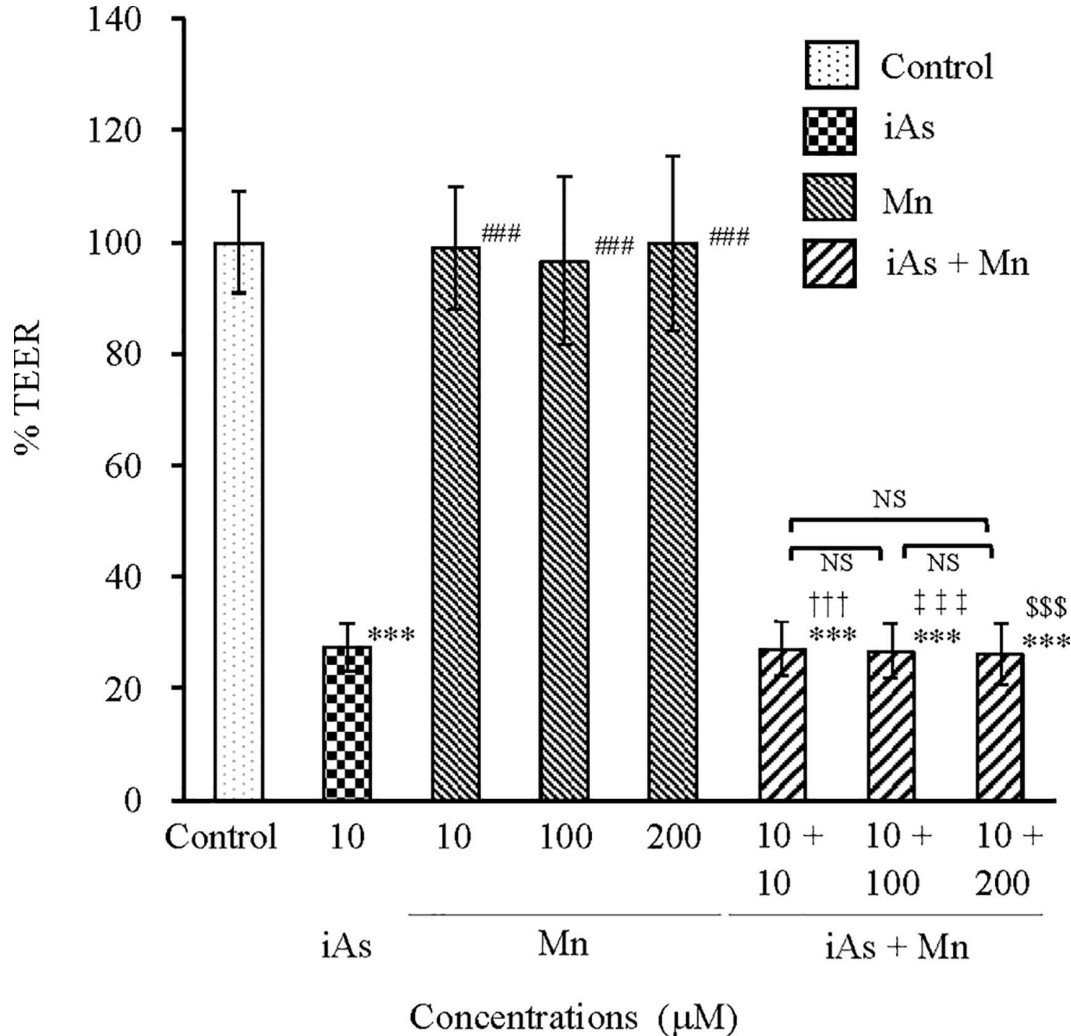

**Fig 6. Changes in TEER after exposure to iAs or Mn alone, and coexposure to iAs and Mn.** TEER was measured at 24 h after exposure to iAs alone (10 μM), Mn alone (10, 100, and 200 μM), and coexposure (iAs + Mn: 10 + 10, 100, and 200 μM). The %TEER values were calculated by normalizing each TEER measurement to the mean TEER value of the control group, which was defined as 100%. Results are expressed as mean ± SD (n = 4). Comparison of the control group, iAs and Mn alone groups, and iAs and Mn coexposure groups was performed using one-way ANOVA with Tukey's post hoc tests. The levels of statistical significance were as follows: for control, ***p < 0.001; for iAs, ###p < 0.001. The significance levels for each concentration of Mn and the corresponding coexposures were as follows: 10 μM, †††p < 0.001; 100 μM, ‡‡‡p < 0.001; 200 μM, $$$p < 0.001. NS indicates no significant differences among the coexposure groups. Detailed statistical values are provided in S6 Table, corresponding to Fig 6.

The results of claudin-5 and ZO-1 expression clearly demonstrated that iAs exposure, and not Mn exposure, caused TJ injury. We speculated that the TJ injury observed in the iAs and Mn coexposure groups was caused by iAs. In other words, Mn did not affect TJ injury due to iAs exposure. The results of %TEER values (Fig 6; see S6 Table) also support this conclusion.

## Discussion

Researchers widely recognize the importance of clarifying the biological effects of combined exposure to harmful substances. Epidemiological data suggest that combined exposure to iAs and Mn worsens cognitive dysfunction [36–38].

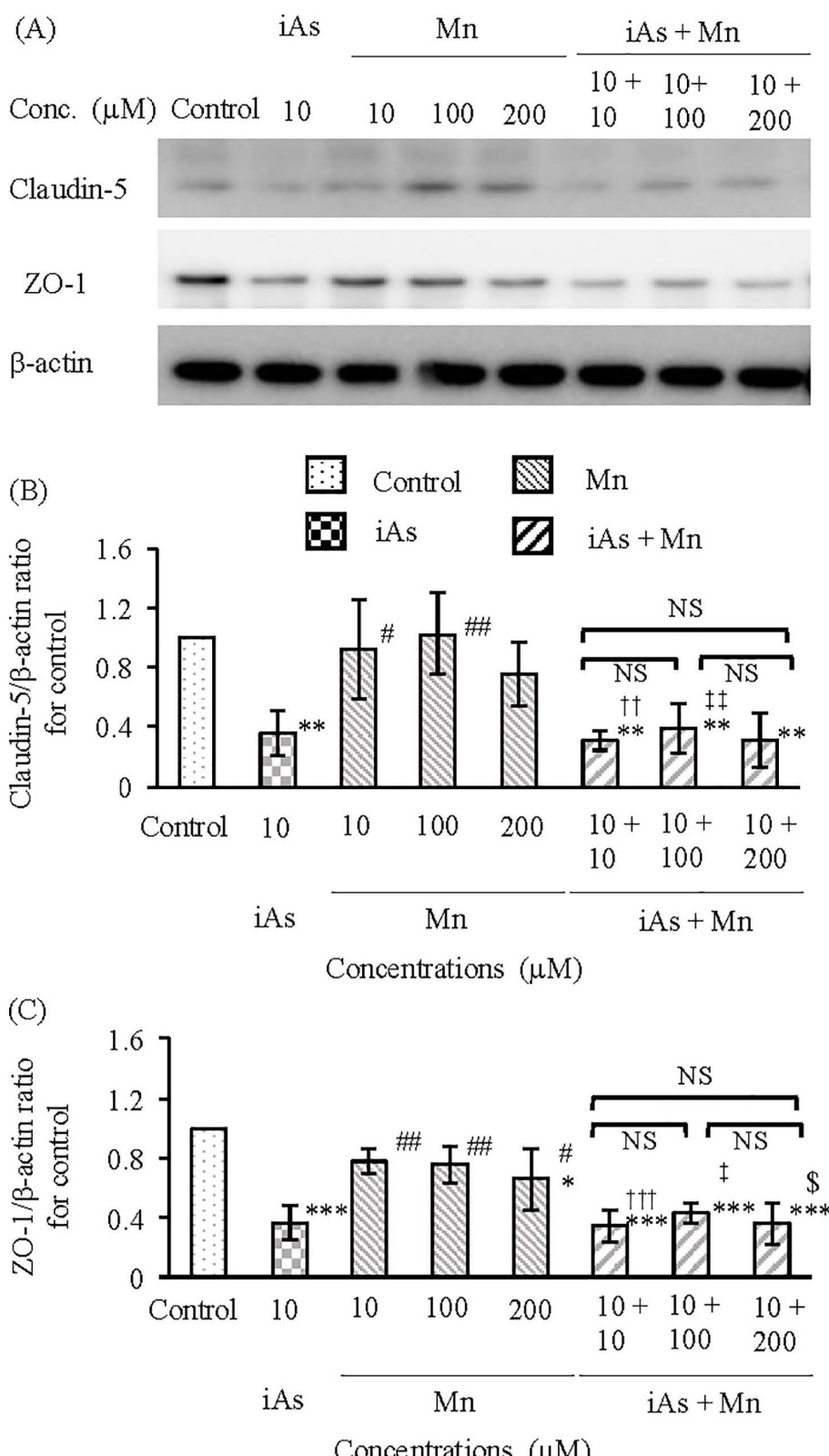

**Fig 7. Expression of the tight junction proteins claudin-5 and ZO-1 after exposure to iAs or Mn alone and coexposure to iAs and Mn. (A):**
Claudin-5 and ZO-1 expressions were measured by WB analysis after 24 h of exposure to iAs alone (10 µM) or Mn alone (10, 100, 200 µM) or iAs and

Mn coexposure (iAs + Mn: 10 + 10, 100, and 200 μM). **(B and C)**: Expression levels of claudin-5 (B) and ZO-1 (C) were quantified by densitometry, standardized to β-actin, and compared with the control value of 1. Results are expressed as mean ± SD (n = 4). Comparison of the control group, iAs and Mn alone groups, and iAs and Mn coexposure groups was performed using one-way ANOVA with Tukey's post hoc tests. The levels of statistical significance were as follows: for control, $^*p < 0.05$, $^{**}p < 0.01$, $^{***}p < 0.001$; for iAs, $^\#p < 0.05$, $^{\#\#}p < 0.01$. The significance levels for each concentration of Mn and the corresponding coexposures were as follows: 10 μM, $^{\dagger\dagger}p < 0.01$, $^{\dagger\dagger\dagger}p < 0.001$; 100 μM, $^\ddagger p < 0.05$, $^{\ddagger\ddagger}p < 0.01$; 200 μM, $^\$p < 0.05$. NS indicates no significant differences among the coexposure groups. Detailed statistical values are provided in S7 Table, corresponding to Figs 7B and 7C.

Research in this field is sparse, and regarding neurotoxicity, there have been studies on the effects of iAs and Mn [52], iAs and Pb [53], iAs, Mn, and Pb [54], and the effects of iAs, Mn, and Pb on motor activity [55], all of which are animal experiments using mice and rats. The joint effects of iAs and other elements suggest antagonistic, additive, or synergistic interactions; however, no definitive conclusion has been reached. Animal experiments are essential; however, there are difficulties involved in analyzing the results. Moreover, when examining the joint effects between toxic elements, the results become more complex when the three elements are targeted. Hence, we recognize the importance of obtaining highly reliable results for the two elements.

### Cytotoxicity, oxidative stress, and inflammatory cytokines in glial cells coexposed to iAs and Mn

**Cytotoxicity of coexposure to iAs and Mn.** We performed cytotoxicity assays using glial cells under exposure to iAs or Mn alone and coexposure to iAs and Mn. As depicted in Fig 1, the cytotoxicity of 5μM-iAs and 10μM-Mn was comparable, with no statistically significant difference. Notably, coexposure to iAs and Mn resulted in greater cytotoxicity than exposure to either iAs or Mn alone, suggesting an additive effect. Furthermore, the magnitude of this additive effect appeared to vary slightly depending on the Mn concentration. This study demonstrated an increase in toxicity of the combined exposure to iAs and Mn. These findings suggest that conventional methods and analysis are insufficient in epidemiological studies of chronic iAs poisoning and chronic Mn poisoning and that new methods need to be developed.

**Oxidative stress in coexposure to iAs and Mn.** Studies have demonstrated that iAs exerts a potent oxidative stress effect, and the antioxidant stress factors Nrf2 and HO-1 are concomitantly expressed [56,57]. In contrast, studies assessing manganese-induced oxidative stress utilizing Nrf2 or HO-1 expression about manganese exposure and oxidative stress are generally less numerous than arsenic studies. Although oxidative stress may play a critical role, it may not be the initiating event [28]. Fig 2 shows the results of Nrf2 and HO-1 mRNA expression 24 h after exposure to iAs, Mn, or their combination in glial cells. The results of Nrf2 mRNA and HO-1 mRNA expression after iAs exposure revealed a substantial increase compared with that in the control group ($p < 0.001$). Conversely, Mn exposure did not alter the levels of Nrf2 mRNA and HO-1 mRNA expression compared with those in the control group. These results indicate that Mn-induced oxidative stress is significantly less pronounced than iAs-induced oxidative stress. Nevertheless, when Mn, which induces only mild oxidative stress, is coexposed with iAs, a notable enhancement in oxidative response is observed. Coexposure to iAs and Mn resulted in increased expression of HO-1 mRNA compared with iAs exposure alone (Fig 2B). This finding suggests that Mn may enhance iAs-induced oxidative stress through additive or synergistic effects. Although the underlying mechanism remains unclear, further investigation is warranted.

It is widely recognized that the Nrf2/HO-1 pathway plays a vital role in combating oxidative stress. Consequently, as shown in Table 1, we attempted to evaluate the intensity of oxidative stress based on the correlation coefficient between the expression levels of Nrf2 mRNA and HO-1 mRNA. The intensity of oxidative stress induced by Mn ($\rho = 0.635$, $p < 0.01$) was lower than that induced by iAs ($\rho = 0.929$, $p < 0.001$). However, the iAs and Mn coexposure group exhibited a correlation coefficient of 0.809 ($p < 0.001$). These results suggest that Mn itself has low oxidative stress effects, but that it enhances the oxidative stress caused by iAs.

**Inflammatory cytokines in coexposure to iAs and Mn.** The amount of information on inflammatory cytokines induced by exposure to iAs or Mn and cognitive dysfunction is limited. In the case of iAs exposure, increased expression

(A)

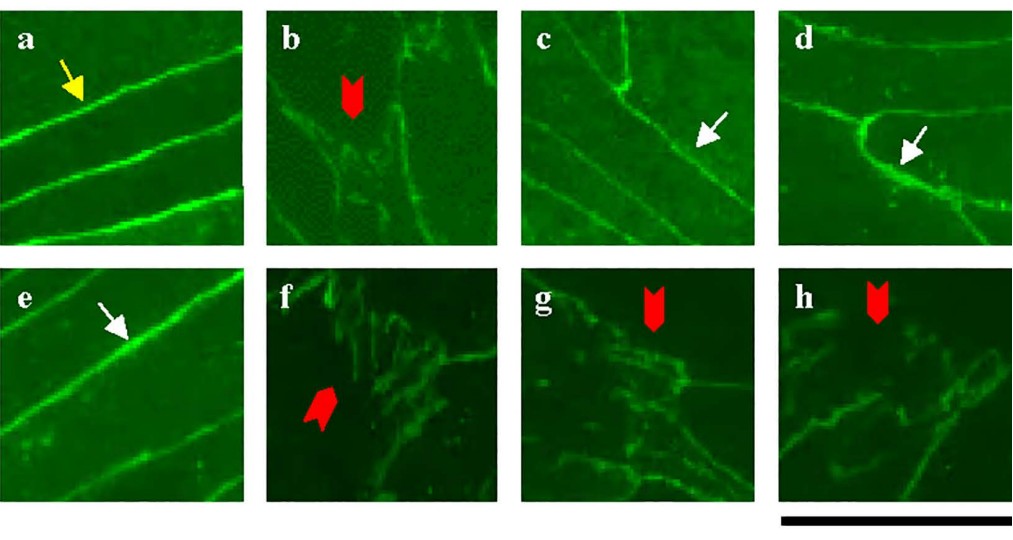

(B)

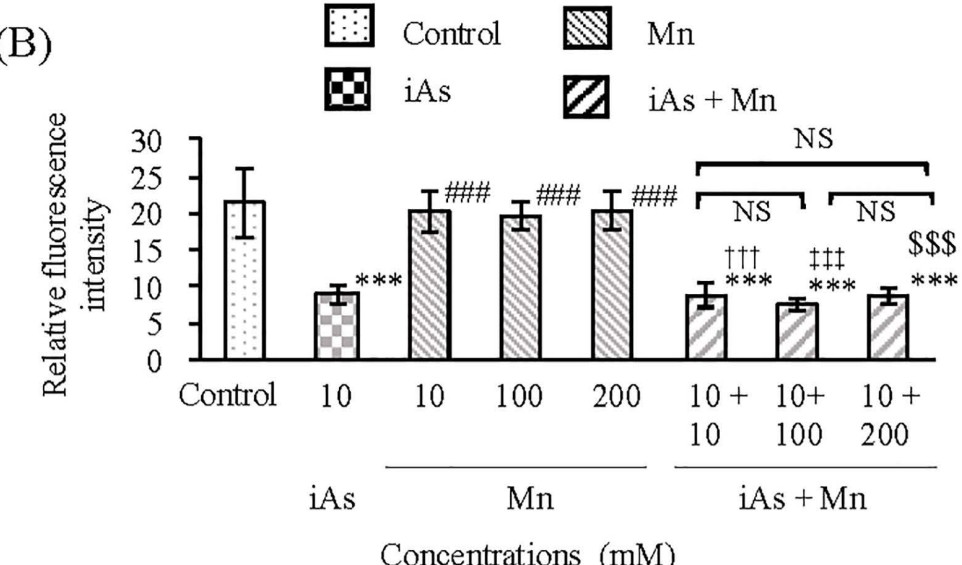

**Fig 8. Altered localization and expression of the tight junction protein claudin-5 by iAs or Mn alone and coexposure to iAs and Mn. (A)**: Representative immunofluorescence staining of claudin-5 (green, Alexa Fluor 488) after 24h exposure: a, control; b, iAs (10 μM); c-e, Mn groups (10, 100, and 200 μM); f-h, iAs and Mn coexposure groups (iAs + Mn: 10 + 10, 100, and 200 μM). The yellow arrow indicates typical TJ protein localization in the control group. Red arrowheads indicate altered TJ protein localization due to iAs-induced damage. White arrows indicate typical TJ protein localization after Mn exposure. Scale bars: 20 μm. **(B)**: Quantitative analysis of the immunofluorescence staining of claudin-5. Relative fluorescence intensity of claudin-5 is expressed as mean±SD (n=6). Comparison of the control group, iAs and Mn alone groups, and iAs and Mn coexposure groups was performed using one-way ANOVA with Tukey's post hoc tests. The levels of statistical significance were as follows: for control, ***$p < 0.001$; for iAs, ###$p < 0.001$. The significance levels for each concentration of Mn and the corresponding coexposures were as follows: 10 μM, †††$p < 0.001$; 100 μM, ‡‡‡$p < 0.001$; 200 μM, $$$$p < 0.001$. NS indicates no significant differences among the coexposure groups. Detailed statistical values are provided in S8 Table, corresponding to Fig 8B.

(A)

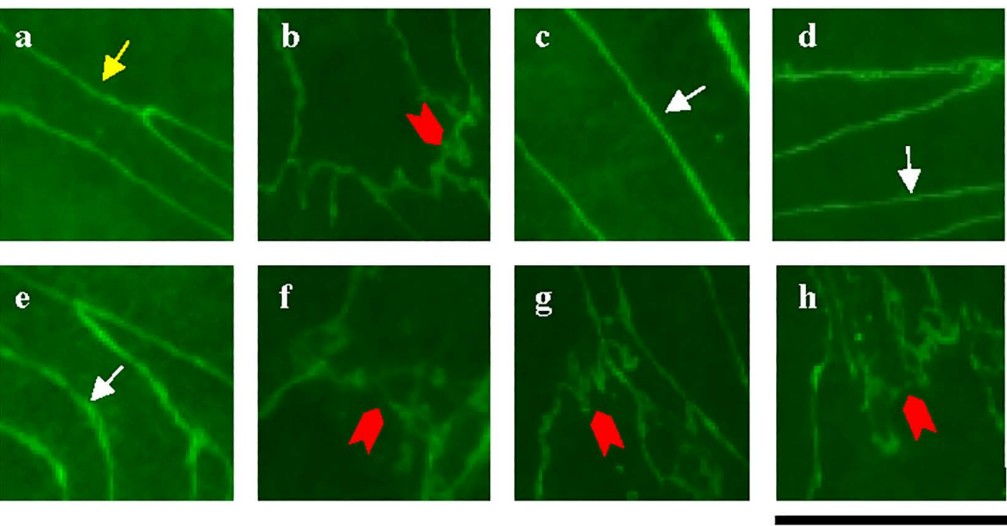

(B)

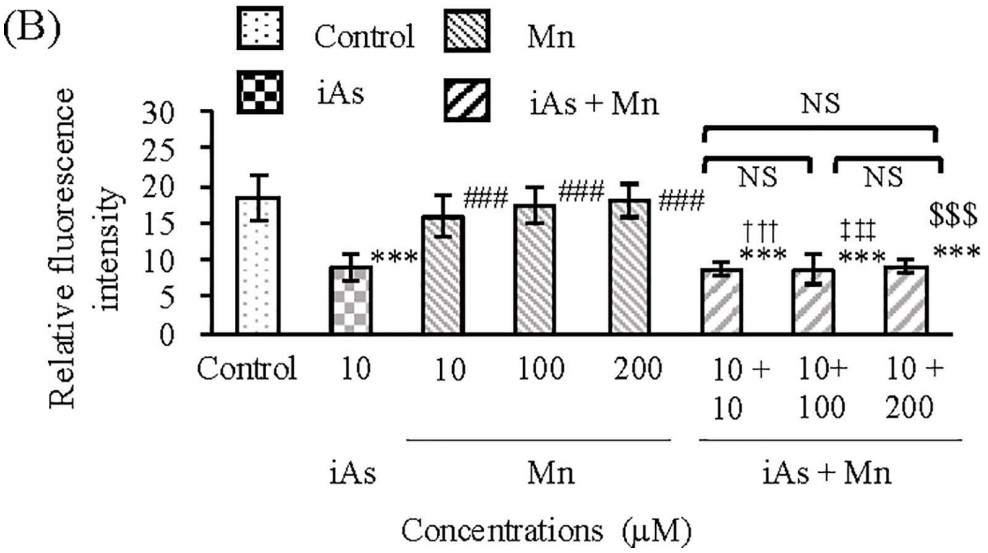

**Fig 9. Altered localization and expression of the tight junction protein ZO-1 by iAs or Mn alone and coexposure iAs and Mn. (A)**: Representative immunofluorescence staining of ZO-1 (green, Alexa Fluor 488) after 24h exposure: a, control; b, iAs (10 μM); c-e, Mn groups (10, 100, and 200 μM); f-h, iAs and Mn coexposure groups (iAs + Mn: 10 + 10, 100, and 200 μM). The yellow arrow indicates typical TJ protein localization in the control group. Red arrowheads indicate altered TJ protein localization due to iAs-induced damage. White arrows indicate typical TJ protein localization after Mn exposure. Scale bars: 20 μm. **(B)**: Quantitative analysis of the immunofluorescence staining of ZO-1. Relative fluorescence intensity of ZO-1 is expressed as mean ± SD (n = 6). Comparison of the control group, iAs and Mn alone groups, and iAs and Mn coexposure groups was performed using one-way ANOVA with Tukey's post hoc tests. The levels of statistical significance were as follows: for control, ***$p < 0.001$; for iAs, ###$p < 0.001$. The significance levels for each concentration of Mn and the corresponding coexposures were as follows: 10 μM, †††$p < 0.001$; 100 μM, ‡‡‡$p < 0.001$; 200 μM, $$$$p < 0.001$. NS indicates no significant differences among the coexposure groups. Detailed statistical values are provided in S9 Table, corresponding to Fig 9B.

of IL-1β [58] and IL-6 [59] has been reported. With Mn exposure, increased expression of IL-1β has been observed [60]. In general, MCP-1 induces IL-1β and IL-6 [61]. As illustrated in Fig 3, exposure to iAs increased the MCP-1 mRNA expression, whereas this result was not observed in the Mn group. Subsequently, we confirmed the expression of IL-1β and IL-6 mRNA. Although there was no significant IL-1β mRNA expression in the iAs or Mn groups, there was an expression at the high concentration of 200 μM Mn. In contrast, there was a statistically significant expression of IL-6 mRNA in the iAs or Mn groups, similar to previous reports [59]. In the group exposed to the iAs and Mn combination, the expression of IL-6 mRNA and IL-1β mRNA (only at 5 μM iAs and 200 μM Mn) increased in conjunction with MCP-1 mRNA expression, indicating a characteristic dependent on Mn concentration (Fig 3). MCP-1 and IL-6 mRNA expression demonstrated a significant correlation (Table 1). The mechanism underlying these results remains unclear. In contrast, it has been confirmed that IL-6 levels increase in the brain tissue of mice exposed to a mixture of iAs, Mn, and Pb [55] compared with those observed with exposure to iAs or Mn alone. In this study, coexposure to iAs and Mn tended to increase IL-6 mRNA expression compared with exposure to iAs or Mn alone. Notably, a synergistic effect was clearly observed in the 100 and 200 μM Mn groups (Fig 3C). These results are consistent with findings from *in vivo* experiments in mice [55]. We believe that the role of inflammatory cytokines in brain tissue following coexposure to iAs and various other elements warrants further investigation.

**Oxidative stress and inflammatory cytokines in interactions.** In both individual exposures to iAs or Mn, and particularly in their coexposure, activation of the antioxidant stress pathway Nrf2/HO-1 was significantly increased (Fig 2B; see S2 Table, panel B). Furthermore, a synergistic effect was suggested by the upregulation of IL-6 mRNA expression (Fig 3C; see S3 Table, panel C). While the precise interpretation of this phenomenon remains unclear, several notable findings were obtained. HO-1, which is induced by Nrf2, is known to play a suppressive role against oxidative stress and inflammatory cytokines [62]. In the coexposure group, significant positive correlations were observed between HO-1 expression and the expression of inflammatory cytokines (MCP-1, IL-1β, and IL-6), with correlation coefficients of $\rho = 0.869$, 0.871, and 0.964, respectively (all $p < 0.001$) (Table 1). These findings suggest that oxidative stress and inflammatory cytokines may interact synergistically in glial cells, potentially enhancing their toxic effects. Such coordinated activation in glial cells could exacerbate neuronal damage.

Our results provide experimental evidence supporting a critical issue raised by epidemiological studies in Bangladesh—namely, that coexposure to iAs and Mn may worsen cognitive dysfunction more than iAs exposure alone [36–38]. Although our findings support this possibility, further studies are needed to draw definitive conclusions.

**TJ injury in BBB with coexposure to iAs and Mn.** iAs [41] and Mn [42] are ingested through the daily diet, and some of them accumulate in brain tissue. It has been confirmed that iAs can cross the BBB and be transferred to the brain tissue in experimental animals [44,45], although there is limited research. Studies using a rat *in vitro* BBB model have provided information on the permeability and metabolism of iAs across the BBB [43]. The known transfer routes of Mn to the brain tissue are the BBB [46], the BCB [47], and the olfactory nerve [48]. Based on research results regarding the transfer of iAs and Mn to brain tissue, it is possible to speculate that iAs and Mn share a common mechanism for passing through the BBB. We believe that information on TJ injury in the BBB is essential for research into cognitive dysfunction caused by iAs or Mn exposure. Therefore, our study focused on the joint effects between iAs and Mn in BBB TJ injury, a mechanism that has not yet been investigated.

In the rat *in vitro* BBB model exposed to iAs, the expression of the antioxidant stress factors Nrf2 and HO-1 was measured in the vascular endothelial cell and pericyte groups and astrocytes. Results demonstrated that the levels of Nrf2 and HO-1 expression in astrocytes were higher than those in the vascular endothelial cell and pericyte groups. This observation is consistent with findings from previous studies, further validating the reliability of the experimental approach [43]. To the best of our knowledge, there have been no reports on the behavior of oxidative stress caused by Mn in the vascular endothelial cell + pericyte group and astrocytes using Nrf2 and HO-1 as indicators. In this study, we measured Nrf2 and HO-1 expression levels in response to exposure to low, medium, and high concentrations of Mn

(10, 100, and 200 μM). In both the vascular endothelial cell and pericyte group and the astrocyte group, there was no change in the concentrations of Nrf2 and HO-1 compared with those in the control group (Figs 4 and 5; see S4 and S5 Tables). This result suggests that Mn does not strongly induce oxidative stress within BBB. The mechanism underlying this phenomenon remains to be elucidated. Mn accumulates in brain tissue and has been reported to accumulate specifically in astrocytes [63]. Further research is necessary on the oxidative stress caused by the synergistic effect of iAs and Mn in astrocytes.

The TJ injury induced by iAs exposure was confirmed by the results of TEER measurements (Fig 6; see S6 Table), claudin-5 expression (Figs 7 and 8; see S7 and S8 Tables), and ZO-1 expression (Figs 7 and 9; see S7 and S9 Tables). These results were consistent with those of previous research [43]. Although it is known that Mn accumulates in the brain tissue [30], there is no information to confirm TJ injury in the BBB. In this study, Mn did not induce TJ injury in the BBB and did not exacerbate the TJ injury induced by iAs (Figs 7-9; see S7–S9 Tables). In other words, Mn did not exert any antagonistic or synergistic effects on TJ injury in the BBB caused by iAs. Nonetheless, it is crucial to acknowledge that Mn is an essential element and is consistently present in the brain tissue [30,42]. Hence, it is conceivable that situations arise where elements or substances exhibiting neurotoxicity come into contact with Mn. Therefore, when exploring the neurotoxicity of Mn, it is imperative to investigate the joint effects of Mn and other neurotoxic substances, particularly compared with iAs.

## Conclusion

To date, limited information has been available regarding the effects of individual or combined exposure to iAs and Mn on BBB TJ injury. Our study demonstrated that iAs induces TJ disruption, whereas Mn alone did not exert such effects, nor did it significantly alter iAs-induced TJ injury. Furthermore, through cytotoxicity assays using glial cells and evaluation of antioxidant stress-related factors (Nrf2, HO-1) and pro-inflammatory cytokines, we obtained compelling evidence suggesting the presence of additive or synergistic effects under coexposure to iAs and Mn. Notably, coexposure resulted in greater toxicity than individual exposures. These findings indicate that coexposure to iAs and Mn may pose a higher risk of cognitive dysfunction than exposure to either iAs or Mn alone. Therefore, it is imperative to strengthen environmental risk assessment related to groundwater or dietary exposure, and to advance experimental research to further elucidate the mechanisms underlying these joint toxic effects.

## Supporting information

**S1 Table. Cytotoxicity of iAs or Mn alone and coexposure to iAs and Mn in glial cells.** *One-way ANOVA with Tukey's post hoc tests. This table supports the statistical comparisons shown in Fig 1.
(XLSX)

**S2 Table. Oxidative stress of iAs or Mn alone and coexposure to iAs and Mn in glial cells.** (A) Nrf2 mRNA expression. (B) HO-1 mRNA expression. *One-way ANOVA with Tukey's post hoc tests.This table supports the statistical comparisons shown in Fig 2A and 2B.
(XLSX)

**S3 Table. Inflammatory cytokines in iAs and Mn alone groups and iAs and Mn coexposure groups in glial cells.** (A) MCP-1 mRNA expression. (B) IL-1β mRNA expression. (C) IL-6 mRNA expression. *One-way ANOVA with Tukey's post hoc tests. This table supports the statistical comparisons shown in Fig 3A–C.
(XLSX)

**S4 Table. Expression of Nrf2 and HO-1 proteins in the vascular endothelial cell and pericyte groups after exposure to iAs or Mn alone and coexposure to iAs and Mn.** (A) Nrf2 protein expression. (B) HO-1 protein

expression. *One-way ANOVA with Tukey's post hoc tests. This table supports the statistical comparisons shown in Fig 4B and 4C.
(XLSX)

**S5 Table. Expression of Nrf2 and HO-1 proteins in astrocytes after exposure to iAs or Mn alone and coexposure to iAs and Mn.** (A) Nrf2 protein expression. (B) HO-1 protein expression. *One-way ANOVA with Tukey's post hoc tests. This table supports the statistical comparisons shown in Fig 5B and 5C.
(XLSX)

**S6 Table. Changes in TEER after exposure to iAs or Mn alone, and coexposure to iAs and Mn.** *One-way ANOVA with Tukey's post hoc tests. This table supports the statistical comparisons shown in Fig 6.
(XLSX)

**S7 Table. Expression of the tight junction proteins claudin-5 and ZO-1 after exposure to iAs or Mn alone and coexposure to iAs and Mn.** (A) Claudin-5 protein expression. (B) ZO-1 protein expression. *One-way ANOVA with Tukey's post hoc tests. This table supports the statistical comparisons shown in Fig 7B and 7C.
(XLSX)

**S8 Table. Quantitative analysis of the immunofluorescence staining of claudin-5 after exposure to iAs or Mn alone and coexposure to iAs and Mn.** *One-way ANOVA with Tukey's post hoc tests. This table supports the statistical comparisons shown in Fig 8B.
(XLSX)

**S9 Table. Quantitative analysis of the immunofluorescence staining of ZO-1 after exposure to iAs or Mn alone and coexposure to iAs and Mn.** *One-way ANOVA with Tukey's post hoc tests. This table supports the statistical comparisons shown in Fig 9B.
(XLSX)

## Author contributions

**Conceptualization:** Hiroshi Yamauchi.

**Data curation:** Toshiaki Hitomi, Hiroko Okuda, Ayako Takata, Hiroshi Yamauchi.

**Formal analysis:** Toshiaki Hitomi, Hiroko Okuda, Ayako Takata.

**Funding acquisition:** Toshiaki Hitomi, Ayako Takata.

**Investigation:** Toshiaki Hitomi, Hiroko Okuda, Ayako Takata, Hiroshi Yamauchi.

**Methodology:** Toshiaki Hitomi, Hiroko Okuda, Ayako Takata.

**Project administration:** Hiroshi Yamauchi.

**Resources:** Toshiaki Hitomi.

**Software:** Toshiaki Hitomi, Hiroko Okuda.

**Supervision:** Ayako Takata, Hiroshi Yamauchi.

**Validation:** Toshiaki Hitomi, Hiroko Okuda, Ayako Takata, Hiroshi Yamauchi.

**Visualization:** Toshiaki Hitomi, Hiroko Okuda, Ayako Takata, Hiroshi Yamauchi.

**Writing – original draft:** Hiroshi Yamauchi.

**Writing – review & editing:** Toshiaki Hitomi, Hiroko Okuda, Ayako Takata, Hiroshi Yamauchi.

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
