## [Decision Letter · Decision Letter 0]

6 Jun 2025

Dear Dr. Yamauchi,

Thank you for submitting your manuscript to PLOS ONE. After careful consideration, we feel that it has merit but does not fully meet PLOS ONE’s publication criteria as it currently stands. Therefore, we invite you to submit a revised version of the manuscript that addresses all the points raised during the review process.

We look forward to receiving your revised manuscript.

Kind regards,

Mária A. Deli, M.D., Ph.D.

Academic Editor

PLOS ONE

Journal Requirements:

Reviewers' comments:

Reviewer's Responses to Questions

**Comments to the Author**

1. Is the manuscript technically sound, and do the data support the conclusions?

Reviewer #1: Partly

Reviewer #2: No

2. Has the statistical analysis been performed appropriately and rigorously?

Reviewer #1: Yes

Reviewer #2: Yes

3. Have the authors made all data underlying the findings in their manuscript fully available?

Reviewer #1: Yes

Reviewer #2: No

4. Is the manuscript presented in an intelligible fashion and written in standard English?

Reviewer #1: Yes

Reviewer #2: Yes

Reviewer #1: GENERAL COMMENT

This manuscript addresses an important topic in environmental toxicology: the combined effects of co-exposure to arsenic (As) and manganese (Mn) on blood-brain barrier (BBB) integrity and oxidative stress. Investigating joint toxicity is highly relevant, as humans are often exposed to metal mixtures (for example, arsenic and manganese frequently co-contaminate drinking water sources. While the subject matter is noteworthy, the manuscript requires substantial improvement in justification, methodology improvement and interpretation of data. Provided the authors address the issues outlined below, the study’s publication could be justified.

MAJOR COMMENTS

INTRODUCTION

-The manuscript in its current form suffers from significant conceptual and presentation issues. The Introduction does not clearly justify the rationale for studying As–Mn joint toxicity: it omits key background information (e.g. both metals’ links to neurodegenerative disease) and fails to define the concept of joint toxic action.

-The introduction fails to compellingly justify why examining the joint toxicity of arsenic and manganese is necessary. The authors mention an example of arsenic poisoning in South America, which is not aligned with the study’s context or focus, and thus feels out of place. Instead, the introduction should highlight scenarios where As and Mn co-exposure is a realistic concern (for instance, co-occurrence of As and Mn in groundwater affecting millions) PMID: 32786605

-More importantly, the text does not mention that both arsenic and manganese are individually known neurotoxicants associated with cognitive impairment and even Alzheimer’s disease risk. The authors should include this to strengthen the rationale. As reviewed somewhere else, chronic exposure to either As or Mn can induce oxidative stress and other neuropathological processes implicated in neurodegeneration. PMID: 40278152

-The introduction is ambiguous about the knowledge gap. It also fails to state whether co-exposure to arsenic and manganese leads to additive, synergistic, or antagonistic effects on the BBB and brain, given that each metal alone can disrupt neurological function.

MATERIALS AND METHODS

-The Materials and Methods lack critical details such as the rationale for dose selection and clarity on experimental controls, raising concerns about reproducibility and the environmental relevance of the findings.

The manuscript does not explain how the chosen doses of arsenic and manganese were determined, which is a critical oversight. The authors must clarify whether the concentrations used in vitro (for both single and combined exposures) reflect environmentally relevant levels (e.g., comparable to concentrations found in blood or drinking water of exposed populations) or were chosen based on prior toxicological studies.

RESULTS

-The Results are described without sufficient critical analysis, for instance, manganese alone appears to have no significant effect on oxidative stress, yet the authors proceed to discuss “joint” effects without demonstrating a true interaction.

-The interpretation of the results, specifically regarding the interaction between arsenic and manganese as reported in several paragraphs of the results section, is a point of major concern. It is not scientifically justified to infer a meaningful interaction if one component is essentially inert (at the tested dose) for the measured endpoints. If Mn alone caused no significant change, then any effect observed in the As+Mn co-exposure group is likely driven solely by arsenic.

MINOR COMMENTS

-The reference to “South American arsenic poisoning” in the Introduction appears misaligned with the study’s focus. If the authors are aiming to set a global context, they should either generalize this point or provide a clearer connection. Otherwise, this example could be removed to tighten the narrative.

-As noted above, the authors should insert a brief definition of joint toxic actions (additivity, synergy, antagonism) for clarity

-It’s suggested to mention in the Introduction (or Discussion) that both As and Mn have been implicated as risk factors in neurodegenerative diseases like Alzheimer’s. The authors allude to cognitive effects but don’t specifically cite evidence. Including a sentence such as, “Notably, epidemiological and mechanistic studies have linked chronic arsenic and manganese exposures to............." would help clarify this aspect.

Reviewer #2: The investigation into the combined toxicity of arsenic and manganese presents an interesting and relevant approach. The study effectively demonstrates the cytotoxic effects of the combined exposure. However, several figures indicate minimal differences between the arsenic-only and the combined exposure groups. This suggests that the proposed pathways may not represent the primary mechanisms underlying the observed effects.

**Do you want your identity to be public for this peer review?** For information about this choice, including consent withdrawal, please see our Privacy Policy

Reviewer #1: **Yes: ** Temitope Adebambo

Reviewer #2: No

---

## [Author Response · Author response to Decision Letter 1]

15 Jul 2025

July 15, 2025

To Reviewer 1

Subject: Response to Reviewers - PONE-D-25-22817 - "Interaction of coexposure to inorganic arsenic and manganese: Tight junction injury of the blood–brain barrier and the relationship between oxidative stress and inflammatory cytokines in glial cells.

Dear Reviewer,

Thank you very much for your insightful and constructive comments, which have significantly contributed to improving the clarity, rigor, and overall quality of our manuscript. Below, we provide point-by-point responses and highlight the corresponding revisions in the manuscript.

GENERAL COMMENT

This manuscript addresses an important topic in environmental toxicology: the combined effects of co-exposure to arsenic (As) and manganese (Mn) on blood-brain barrier (BBB) integrity and oxidative stress. Investigating joint toxicity is highly relevant, as humans are often exposed to metal mixtures (for example, arsenic and manganese frequently co-contaminate drinking water sources. While the subject matter is noteworthy, the manuscript requires substantial improvement in justification, methodology improvement and interpretation of data. Provided the authors address the issues outlined below, the study’s publication could be justified.

Thank you for recognizing the importance of our study. We agree that the manuscript required substantial improvements in justification, methodology, and data interpretation. We have carefully considered your suggestions and revised the manuscript accordingly.

MAJOR COMMENTS

INTRODUCTION

We appreciate these crucial comments. We acknowledge the shortcomings in our introduction and have revised it as follows:

Comment -1: The manuscript in its current form suffers from significant conceptual and presentation issues. The Introduction does not clearly justify the rationale for studying As–Mn joint toxicity: it omits key background information (e.g. both metals’ links to neurodegenerative disease) and fails to define the concept of joint toxic action.

Response:

Comment-1: Thank you very much for your valuable comment. We have revised the Introduction section to more clearly justify the rationale for studying the joint toxicity of arsenic (iAs) and manganese (Mn) (lines 52–56).

Specifically, we have added important background information indicating that both iAs and Mn are individually associated with cognitive dysfunction and are implicated in the development of neurodegenerative diseases such as Alzheimer’s disease (lines 58–67).

In accordance with the reviewers' comments and instructions, we have included the definition of “joint effects” in the introduction of this study.

“In this study, we evaluated the joint effects of iAs and Mn by distinguishing additive effects—where the combined outcome equals the sum of individual effects (EiAs + EMn = EiAs + Mn)—from synergistic effects, where the combined outcome exceeds this sum (EiAs + Mn > EiAs + EMn). Here, EiAs denotes the effect of exposure to iAs, EMn the effect of exposure to Mn, and EiAs + Mn the effect of coexposure.”

(lines 109-113).

Furthermore, we emphasized the real-world relevance of such coexposure by referencing environmental contamination cases (e.g., groundwater contamination in Bangladesh; Refs. 36–38), and noted that iAs and Mn can co-occur in both groundwater (Refs. 39, 40) and the diet (Refs. 41, 42), potentially affecting the general population. These revisions can be found in lines 68–85 of the revised manuscript.

Comment -2: The introduction fails to compellingly justify why examining the joint toxicity of arsenic and manganese is necessary. The authors mention an example of arsenic poisoning in South America, which is not aligned with the study’s context or focus, and thus feels out of place. Instead, the introduction should highlight scenarios where As and Mn co-exposure is a realistic concern (for instance, co-occurrence of As and Mn in groundwater affecting millions) PMID: 32786605

Response:

Comment-2:

We appreciate the reviewer’s insightful comment. As suggested, we removed the unrelated information regarding chronic arsenic poisoning, arsenic carcinogenicity, and lifestyle-related diseases, as these were not directly relevant to the scope of our study.

We also revised the introduction to more clearly justify the necessity of investigating the joint toxicity of iAs and Mn.

Specifically, we added information emphasizing realistic and high-impact scenarios where co-exposure to these elements is a serious concern. For instance, we cited a recent report (PMID: 32786605; Ref. 39) documenting the co-occurrence of arsenic and manganese in groundwater in Bangladesh, where millions of people are affected by this contamination.

(lines 75–80.)

This revision enhances the relevance and urgency of our research within the context of environmental health.

Comment -3: More importantly, the text does not mention that both arsenic and manganese are individually known neurotoxicants associated with cognitive impairment and even Alzheimer’s disease risk. The authors should include this to strengthen the rationale. As reviewed somewhere else, chronic exposure to either As or Mn can induce oxidative stress and other neuropathological processes implicated in neurodegeneration. PMID: 40278152

Response:

Comment-3:

We have addressed this comment in lines 62–67. We clearly stated that iAs and Mn are each known to be neurotoxic and are associated with cognitive dysfunction and an increased risk of Alzheimer’s disease. To support this, we cited a recent review (PMID: 40278152; Ref. 33) that discusses neuropathological mechanisms, such as oxidative stress, induced by chronic exposure to these metals. Furthermore, we supplemented this background with evidence from animal studies (lines 56–60) demonstrating cognitive dysfunction following exposure to iAs and Mn.

Comment -4: The introduction is ambiguous about the knowledge gap. It also fails to state whether co-exposure to arsenic and manganese leads to additive, synergistic, or antagonistic effects on the BBB and brain, given that each metal alone can disrupt neurological function.

Response:

Comment-4:

The corresponding revision is found in lines 97–107. We now clarify that while iAs is known to disrupt the BBB through tight junction (TJ) injury, there is limited information on the direct impact of Mn on BBB integrity. Given this knowledge gap, we emphasize the importance of exploring whether coexposure to iAs and Mn produces additive or synergistic effects on TJ injury.

MATERIALS AND METHODS

Comment -5:

The Materials and Methods lack critical details such as the rationale for dose selection and clarity on experimental controls, raising concerns about reproducibility and the environmental relevance of the findings. The manuscript does not explain how the chosen doses of arsenic and manganese were determined, which is a critical oversight. The authors must clarify whether the concentrations used in vitro (for both single and combined exposures) reflect environmentally relevant levels (e.g., comparable to concentrations found in blood or drinking water of exposed populations) or were chosen based on prior toxicological studies.

Response

Thank you for these important comments. To enhance the clarity and reproducibility of the Materials and Methods section, we have made the following changes:

Comment -5:

The exposure concentrations were determined with reference to survey results from Bangladesh. In a survey on cognitive dysfunction in children, the concentrations of arsenic and Mn in groundwater in an area of chronic arsenic poisoning were reported as 117.8 ± 145.2 µg/L and 1,386 ± 927 µg/L, respectively [1]. For reference, 5 µM iAs corresponds to 378 µg/L, and 10 µM Mn corresponds to 549 µg/L. In this experiment, the lowest exposure concentrations (5 µM iAs and 10 µM Mn) were approximately 3-fold higher than the survey value for iAs and approximately 0.4-fold (i.e., 40%) of the survey value for Mn. Therefore, these concentrations were considered to reflect environmentally relevant exposure levels. Furthermore, in the Mn group, concentrations corresponding to 10-fold and 20-fold the measured value of 1,386 µg/L reported in the survey were also applied.

(lines 140–148.)

(Ref. 1) Wasserman GA, Liu X, Parvez F, Ahsan H, Factor-Litvak P, van Geen A, et al. Water arsenic exposure and children’s intellectual function in Araihazar, Bangladesh. Environ Health Perspect. 2004 Apr; 112(13):1329–1333. doi: 10.1289/ehp.6964.

RESULTS

We appreciate these crucial observations. To improve the interpretation and presentation of our results, we have made the following revisions:

Comment -6:

The Results are described without sufficient critical analysis, for instance, manganese alone appears to have no significant effect on oxidative stress, yet the authors proceed to discuss “joint” effects without demonstrating a true interaction.

The interpretation of the results, specifically regarding the interaction between arsenic and manganese as reported in several paragraphs of the results section, is a point of major concern. It is not scientifically justified to infer a meaningful interaction if one component is essentially inert (at the tested dose) for the measured endpoints.

Response

Comment -6:

There was a significant error in the description of the results. Specifically, for some of the iAs + Mn results, although “additive effect” was the appropriate term, it was ambiguously expressed as “synergistic effect.” We have thoroughly reviewed all results and clarified the distinction between additive and synergistic effects for iAs + Mn, revising and adding the relevant text in the results section.

The response is the same as that given in response to major comment-1. We set joint effects in this study as follows.

In accordance with the reviewers' comments and instructions, we have included the definition of “joint effects” in the introduction of this study.

“In this study, we evaluated the joint effects of iAs and Mn by distinguishing additive effects—where the combined outcome equals the sum of individual effects (EiAs + EMn = EiAs + Mn)—from synergistic effects, where the combined outcome exceeds this sum (EiAs + Mn > EiAs + EMn). Here, EiAs denotes the effect of exposure to iAs, EMn the effect of exposure to Mn, and EiAs + Mn the effect of coexposure.”

(lines 109-113).

Additionally, we have revised and added text regarding “joint effects” throughout the results; the revised and newly added portions have been highlighted in yellow in the manuscript for clarity. Additionally, since we have changed the interpretation of additive and synergistic effects in the results, we have revised the relevant sections in the “Discussion,” highlighting the revised and added portions in yellow. Furthermore, we have revised the content of the ‘Abstract’ and ‘Conclusion’ sections to ensure accuracy.

Comment -7:

If Mn alone caused no significant change, then any effect observed in the As+Mn co-exposure group is likely driven solely by arsenic.

Response

Comment -7:

As pointed out by the reviewer, when no significant changes were observed following Mn exposure alone, it is reasonable to assume that specific effects in the coexposure group may primarily reflect the influence of iAs. In our study, this was indeed the case for specific endpoints, such as the expression of Nrf2 and HO-1 in glial cells and BBB-astrocytes, as well as the mRNA levels of MCP-1 and IL-6. However, our data also demonstrated that specific markers—particularly HO-1 and IL-6 in glial cells, and HO-1 in BBB-astrocytes—showed significantly greater responses in the coexposure group than in either of the single exposure groups. These findings suggest synergistic effects between iAs and Mn, despite Mn showing no significant impact when applied alone. Therefore, Mn may act as a potentiating factor under coexposure conditions, enhancing iAs-induced toxicity beyond the level expected from iAs alone. We have clarified this interpretation in the revised manuscript and emphasized that the identification of such interactions is a key contribution of this study to the understanding of joint iAs and Mn toxicity.

MINOR COMMENTS

Comment -8:

The reference to “South American arsenic poisoning” in the Introduction appears misaligned with the study’s focus. If the authors are aiming to set a global context, they should either generalize this point or provide a clearer connection. Otherwise, this example could be removed to tighten the narrative.

Response

Comment -8:

Thank you for your comment. By the reviewers' comments, descriptions of carcinogenicity and lifestyle-related diseases associated with chronic arsenic poisoning have been deleted. The occurrence of cognitive dysfunction due to exposure to iAs and Mn alone has been reorganized and described.

lines 52-56.

Comment -9:

As noted above, the authors should insert a brief definition of joint toxic actions (additivity, synergy, antagonism) for clarity.

Response

Comment -9:

Thank you for your comment. The response to this comment is described in major comment -1 and -6. We have addressed this comment in lines 109–113.

Comment -10:

It’s suggested to mention in the Introduction (or Discussion) that both As and Mn have been implicated as risk factors in neurodegenerative diseases like Alzheimer’s. The authors allude to cognitive effects but don’t specifically cite evidence. Including a sentence such as, “Notably, epidemiological and mechanistic studies have linked chronic arsenic and manganese exposures to...........” would help clarify this aspect.

Response

Comment -10:

Thank you for your comment. The response to this comment is described in major comment -3.

We have addressed this comment in lines 62–67. We clearly stated that iAs and Mn are each known to be neurotoxic and are associated with cognitive dysfunction and an increased risk of Alzheimer’s disease. To support this, we cited a recent review (PMID: 40278152; Ref. 33) that discusses neuropathological mechanisms, such as oxidative stress, induced by chronic exposure to these metals.

To Reviewer 2

Comments

The investigation into the combined toxicity of arsenic and manganese presents an interesting and relevant approach. The study effectively demonstrates the cytotoxic effects of the combined exposure. However, several figures indicate minimal differences between the arsenic-only and the combined exposure groups. This suggests that the proposed pathways may not represent the primary mechanisms underlying the observed effects.

Response

Thank you for acknowledging the relevance of our study on the combined toxicity of arsenic and manganese and for appreciating that our study effectively demonstrates the cytotoxic effects of the combined exposure.

We revised the “Results” and “Discussion” sections of the manuscript to provide a more detailed and careful interpretation of the data. Specifically, we supplemented the explanation of the potential additive and synergistic effects of iAs and Mn in coexposure. Information in this field is limited, and we have discussed the possible mechanisms as thoroughly as possible; however, further research is still needed. The revised and newly added sections are highlighted in yellow for clarity in the “Revised Article with Changes Highlighted” file.

Thank you again for your time and effort in reviewing our manuscript. Your feedback has significantly helped us to improve this study.

Hiroshi Yamauchi, Ph. D.

Visiting Professor

Department of Preventive Medicine, St. Marianna University School of Medicine, Japan

Email: hyama@marianna-u.ac.jp

Telephone: +81-44-978-6392

Fax: +81-44-978-6393

---

## [Decision Letter · Decision Letter 1]

30 Jul 2025

Interaction of coexposure to inorganic arsenic and manganese: Tight junction injury of the blood–brain barrier and the relationship between oxidative stress and inflammatory cytokines in glial cells

PONE-D-25-22817R1

Dear Dr. Yamauchi,

We’re pleased to inform you that your manuscript has been judged scientifically suitable for publication and will be formally accepted for publication once it meets all outstanding technical requirements.

Kind regards,

Mária A. Deli, M.D., Ph.D.

Academic Editor

PLOS ONE

Additional Editor Comments (optional):

Reviewers' comments:

Reviewer's Responses to Questions

**Comments to the Author**

Reviewer #1: (No Response)

2. Is the manuscript technically sound, and do the data support the conclusions?

Reviewer #1: Yes

3. Has the statistical analysis been performed appropriately and rigorously?

Reviewer #1: Yes

4. Have the authors made all data underlying the findings in their manuscript fully available?

Reviewer #1: Yes

5. Is the manuscript presented in an intelligible fashion and written in standard English?

Reviewer #1: Yes

Reviewer #1: Authors have addressed the concerns I raised in the first round of reviews. The updated report now includes how Mn and Arsenic exposures are individually linked to neurodegenerative diseases.

The real-worlds relevance has also been highlighted and the ambiguity in the introduction has been addressed.

**Do you want your identity to be public for this peer review?** For information about this choice, including consent withdrawal, please see our Privacy Policy

Reviewer #1: **Yes: ** Temitope H Adebambo

---

## [Editor Report · Acceptance letter]

PONE-D-25-22817R1

PLOS ONE

Dear Dr. Yamauchi,

I'm pleased to inform you that your manuscript has been deemed suitable for publication in PLOS ONE. Congratulations! Your manuscript is now being handed over to our production team.

Kind regards,

on behalf of

Prof. Mária A. Deli

Academic Editor

PLOS ONE